# Omicron Spike confers enhanced infectivity and interferon resistance to SARS-CoV-2 in human nasal tissue

Guoli Shi[1,3], Tiansheng Li[2,3], Kin Kui Lai [1], Reed F. Johnson[2], Jonathan W. Yewdell[2] & Alex A. Compton [1] ✉

Omicron emerged following COVID-19 vaccination campaigns, displaced previous SARS-CoV-2 variants of concern worldwide, and gave rise to lineages that continue to spread. Here, we show that Omicron exhibits increased infectivity in primary adult upper airway tissue relative to Delta. Using recombinant forms of SARS-CoV-2 and nasal epithelial cells cultured at the liquid-air interface, we show that mutations unique to Omicron Spike enable enhanced entry into nasal tissue. Unlike earlier variants of SARS-CoV-2, our findings suggest that Omicron enters nasal cells independently of serine transmembrane proteases and instead relies upon metalloproteinases to catalyze membrane fusion. Furthermore, we demonstrate that this entry pathway unlocked by Omicron Spike enables evasion from constitutive and interferon-induced antiviral factors that restrict SARS-CoV-2 entry following attachment. Therefore, the increased transmissibility exhibited by Omicron in humans may be attributed not only to its evasion of vaccine-elicited adaptive immunity, but also to its superior invasion of nasal epithelia and resistance to the cell-intrinsic barriers present therein.

Efficient infection of human cells by severe acute respiratory syndrome coronavirus 2 (SARS-CoV-2) requires the interaction of SARS-CoV-2 Spike with its receptor at the cell surface, angiotensin-converting enzyme 2 (ACE2)[1,2]. In addition, Spike has been described to bind to various cellular factors to promote coronavirus attachment to the cell surface. 78-kD glucose-regulated protein, neuropilin-1, high-density lipoprotein scavenger receptor B type 1, CD209/CD299, Axl, sialic acid, and heparan sulfate have all been reported to interact with Spike and promote the virus entry process[3-9]. In general, it is thought that factors aiding virus attachment enable subsequent ACE2 binding. Engagement of ACE2 alters Spike conformation and facilitates its processing by cellular proteases, such as serine transmembrane proteases like TMPRSS2, matrix metalloproteinases (MMP), or a disintegrin and metalloproteinases (ADAM)[2,10-14]. Protease cleavage enables Spike to trigger fusion between viral and cellular membranes to complete cellular entry[15].

The Omicron B.1.1.529.1 (BA.1) variant emerged in late 2021 and, as a result of increased transmissibility between humans[16,17], quickly replaced the Delta variant of concern as the dominant form of SARS-CoV-2 circulating worldwide (https://www.cdc.gov/coronavirus/2019-ncov/variants/variant-classifications.html). Compared to previous SARS-CoV-2 variants of concern, Spike from BA.1 contained a plethora of unique mutations, and subsequent Omicron lineages (including the BA.2, BA.4, BA.5, BQ.1, and XBB subvariants) contained additional mutations in Spike (https://covid.cdc.gov/covid-data-tracker/#datatracker-home). The functional characterization of Omicron has revealed that said Spike mutations, including those found in the receptor binding domain, allow for evasion of neutralizing antibodies induced by COVID-19 vaccination[18-26]. Furthermore, some studies have shown that Omicron Spike displays an increased affinity for human ACE2 compared to the early D614G variant[23,27,28], but this was not supported by others[29]. Therefore, it remains unclear whether antibody

[1]HIV Dynamics and Replication Program, Center for Cancer Research, National Cancer Institute, Frederick, MD, USA. [2]Laboratory of Viral Diseases, National Institute of Allergy and Infectious Diseases, Bethesda, MD, USA. [3]These authors contributed equally: Guoli Shi, Tiansheng Li. ✉e-mail: alex.compton@nih.gov

evasion and ACE2 affinity are the only factors explaining the improved transmissibility of Omicron subvariants.

It has also been reported that Omicron utilizes a distinct cellular entry pathway compared to ancestral or early forms of SARS-CoV-2[30–34]. Specifically, the proteolytic activation of Omicron Spike is less dependent on TMPRSS2 but may be more dependent on cathepsin activity in endolysosomes or metalloproteinase activity at the plasma membrane[11]. However, the precise characterization of the entry pathways used by Omicron in physiologically relevant primary cells is lacking, and whether they are sensitive to cell-intrinsic antiviral barriers has not been determined. The apparent loss of TMPRSS2 dependence may have consequences for the cellular tropism of Omicron in vivo. Some reports suggest that Omicron BA.1 exhibits a decreased propensity for lower respiratory tract (lung) infection, which may explain the decreased pathogenicity associated with Omicron infections[35–40]. Since the upper respiratory tract, and specifically, the nasal epithelium, represents the initial site of SARS-CoV-2 infection and the initial protective innate immune response[41,42], we compared the ability of Omicron to invade human nasal epithelia relative to ancestral SARS-CoV-2 and the preceding variant of concern, Delta, using a primary human nasal tissue culture model. We found that Omicron BA.1 and BA.2 exhibited a markedly enhanced infectivity in primary nasal epithelia compared to early SARS-CoV-2, and using a recombinant virus, we show that Spike from Omicron, but not Delta, controlled this phenotype. Furthermore, despite Omicron eliciting a type-I interferon response in nasal tissue, Omicron Spike used a metalloproteinase-mediated entry pathway that was associated with evasion of interferon-inducible factors targeting virus entry. Collectively, these findings may explain the increased transmissibility of Omicron as well as its ability to displace more interferon-sensitive variants of SARS-CoV-2.

Overall, our findings suggest that the global spread of Omicron was fueled not only by increased ACE2 affinity and resistance to vaccine-induced adaptive immunity but also by resistance to nasal cell-intrinsic entry inhibitors.

## Results

### Omicron BA.1 exhibits superior replicative fitness relative to early SARS-CoV-2 and Delta in primary human nasal epithelial cells

Primary human nasal epithelial cells isolated from three donors were pooled and cultured as submerged monolayers, which allows for the propagation of cells in the basal (undifferentiated) state. Immunofluorescence staining of acetylated tubulin indicated that mature cilia are lacking in cells cultured in this manner, although they express detectable levels of ACE2 intracellularly and at or near the plasma membrane (Fig. 1A and Supplemental Fig. 1). Nasal monolayers were inoculated with early/ancestral SARS-CoV-2 (USA-WA1/2020 strain, herein referred to as WA1) at a multiplicity of infection (MOI) of 0.05, and RT-qPCR of viral ORF1a RNA at multiple time points was suggestive of virus replication over time (Fig. 1B). Productive infection was confirmed by demonstrating that cell culture supernatants contained infectious virus that could be titered on Vero E6-ACE2-TMPRSS2 cells by ORF1a RT-qPCR and anti-nucleocapsid immunostaining (Supplemental Fig. 2A). By comparison, inoculation of nasal monolayers with the same MOI of BA.1 resulted in far greater virus replication at all time points (Fig. 1B). In contrast, BA.1 did not replicate in submerged monolayers of human small airway (lung) epithelial cells (three pooled donors), while WA1 replicated with similar kinetics in nasal and lung cells (Fig. 1C). These data suggest that BA.1 displays a growth advantage specific to the upper airway of the human respiratory tract. BA.1 exhibited an approximately 100-fold greater replication capacity in nasal monolayers compared to WA1, either when measuring absolute (Supplemental Fig. 2B) or relative (Supplemental Fig. 2C) copy numbers of viral ORF1a. We found that this replication advantage by BA.1

was accompanied by an approximately 30-fold superior capacity to adhere to the nasal cell surface compared to WA1 (Supplemental Fig. 2D). Since the concentrations of viral RNA varied between our virus stocks (Supplemental Table 1), we excluded that WA1 and BA.1 inoculants (initially titered on Vero E6-ACE2-TMPRSS2 cells) contained different quantities of virus particles by inoculating HEK293T-ACE2 and Vero E6-ACE2-TMPRSS2 cells. In contrast to our observations in nasal monolayers, WA1 and BA.1 infected these permissive cell lines to equal extents (Supplemental Fig. 2E, F). Even when virus inoculants were calculated using an alternative virus titering strategy (based on viral ORF1a RT-qPCR instead of Vero E6-ACE2-TMPRSS2 infection (Supplemental Table 1), BA.1 maintained a significant replicative advantage in nasal monolayers relative to WA1 (Supplemental Fig. 2G). Therefore, BA.1 exhibits enhanced replicative potential in primary human nasal cells relative to WA1, and this is associated with improved cell surface adhesion by BA.1 Spike. Since BA.1 emerged in humans while the preceding variant of concern, Delta, was widespread, we compared the capacity for BA.1 and Delta to infect human nasal monolayers. In contrast to the orders of magnitude improvement exhibited by BA.1, Delta infected nasal cells only modestly better than WA1, and the difference was not statistically significant (Fig. 1D). Therefore, the enhanced ability for BA.1 to infect primary human nasal epithelia emerged in the Omicron ancestor, not in Delta. Importantly, remdesivir, an inhibitor of the viral RNA polymerase, significantly reduced ORF1a levels in nasal cells inoculated with WA1, Delta, and BA.1, demonstrating that measurement of ORF1a by RT-qPCR is reflective of virus replication (Fig. 1E).

During the preparation of this manuscript, it was reported that Omicron displays greater adhesion to human nasal cells, and this was attributed to enhanced binding to cilia in differentiated nasal epithelia[43]. Our results here suggest that, besides cilia, other factors intrinsic to nasal cells must govern the improved replicative fitness of BA.1 compared to early SARS-CoV-2.

### Omicron Spike enables enhanced SARS-CoV-2 infectivity in primary nasal epithelia cultured at the air–liquid interface

To recreate the three-dimensional, pseudostratified architecture of nasal epithelia in vivo, primary human nasal epithelial cells were pooled from 14 donors and differentiated at the air–liquid interface (ALI). In addition to ample ACE2 expression, differentiation status was confirmed by stratification of nuclei and the presence of mature cilia at the tissue surface (Fig. 2A). Relative to nasal monolayers (Supplemental Fig. 2B), nasal ALI were more permissive to SARS-CoV-2 infection, as assessed by absolute ORF1a copy number (Supplemental Fig. 3A). In this tissue, replication of Omicron BA.1 and BA.2 exceeded that of WA1 by multiple orders of magnitude at 48 h post-inoculation, with BA.2 exhibiting the greatest replicative potential (Fig. 2B). To confirm that nasal ALI support repeated rounds of productive infection, we collected culture medium supernatants of nasal ALI inoculated with WA1, BA.1, or BA.2 and quantified infectious virus levels by titrating on Vero E6-ACE2-TMPRSS2 cells. In agreement with RT-qPCR results, BA.1 and BA.2 achieved higher infectious titers in nasal ALI relative to WA1 (Fig. 2C). In accordance with enhanced virus replication of BA.1 and BA.2, we detected elevated type-I interferon induction (*IFNB*) at 48 h post-inoculation (Fig. 2D).

Given the degree to which Omicron BA.1 and BA.2 replicate in primary nasal ALI, we also tested the infective potential of an Omicron sublineage more recently circulating in humans (BQ.1) as well as Delta and the D614G variant, the ancestor of Pango lineage B that gave rise to all SARS-CoV-2 variants of concern. D614G replicated to a similar extent as WA1, while Delta achieved a significantly higher degree of replication (23-fold). However, replication of Omicron lineages BA.1 and BQ.1 in nasal ALI far exceeded that of all variants tested (282- and 88-fold higher than WA1, respectively) (Fig. 2E). While all viruses triggered the expression of the interferon-stimulated gene *IFITM3*

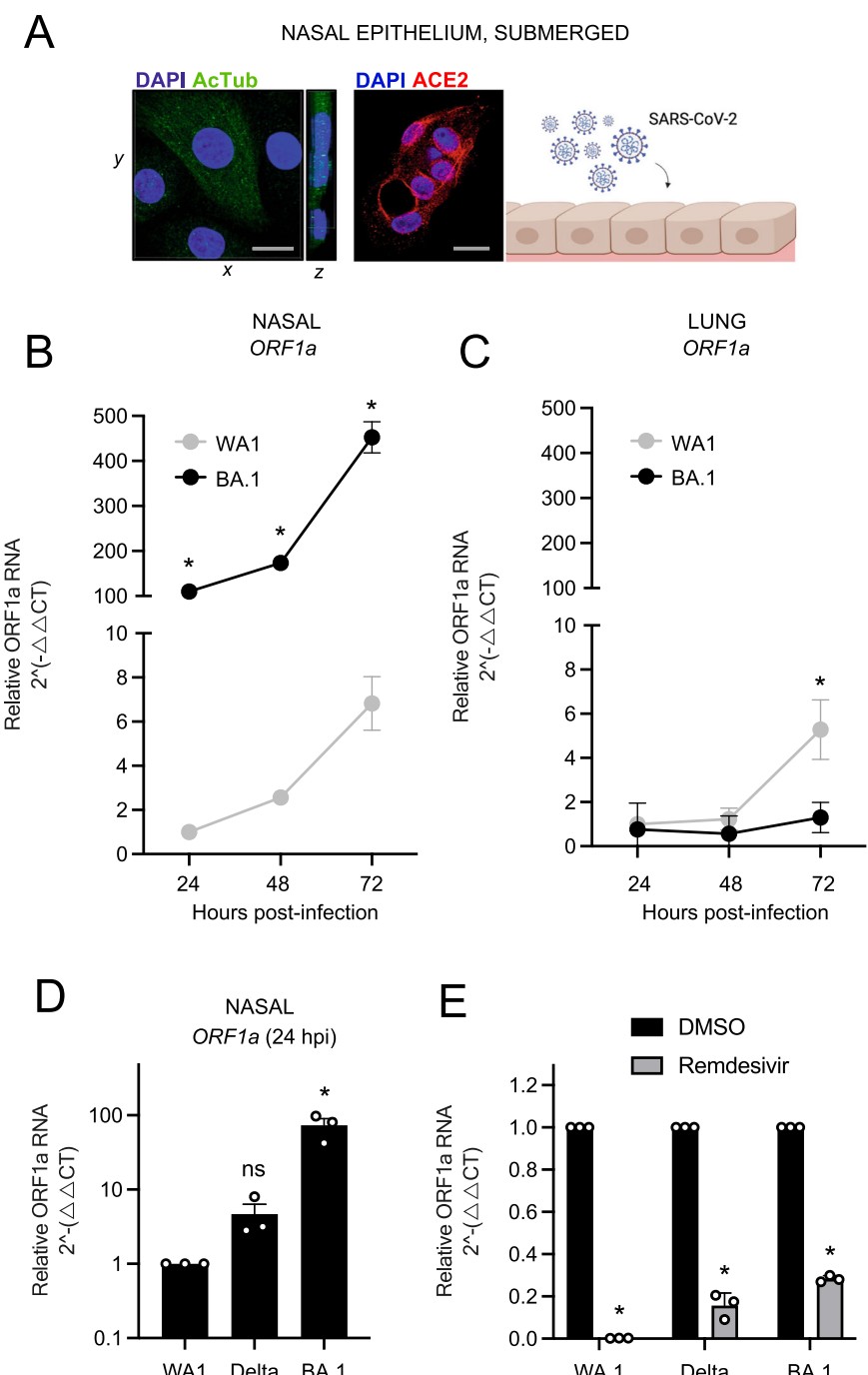

**Fig. 1 | Omicron BA.1 exhibits superior replicative fitness relative to early SARS-CoV-2 and Delta in primary human nasal epithelial cells. A** Primary human nasal epithelial cells (pooled from three human donors) were cultured as undifferentiated, submerged monolayers and challenged with SARS-CoV-2. Cells were fixed and permeabilized for confocal immunofluorescence microscopy. Acetylated tubulin levels were determined by anti-AcTub immunofluorescence, ACE2 levels were determined by anti-ACE2 immunofluorescence, and DAPI was used to stain nuclei. 3D reconstructions of *xy* and *yz* fields are shown. Scale bars = 10 μm. Cartoon made with Biorender.com. **B** Primary human nasal epithelial cells (pooled from three donors) or **C** human small airway (lung) epithelial cells (pooled from three donors) were challenged with replication-competent SARS-CoV-2 WA1 or Omicron BA.1 at an MOI of 0.05. Total cellular RNA was extracted, and viral ORF1a was quantified by RT-qPCR at the indicated time points. Relative viral RNA abundance compared to actin was determined by the $2^{(-\Delta\Delta CT)}$ method. ORF1a abundance of WA1 at 24 h post inoculation was set to 1. **D** Virus replication in primary human nasal epithelial cells was measured by detecting ORF1a levels with RT-qPCR at 24 h post inoculation with WA1, Delta, or BA.1 at an MOI of 0.05. **E** As in **D**, except that primary human nasal epithelial cells were treated with remdesivir (10 μM) or DMSO prior to and during inoculation with WA1, Delta, or BA.1. ORF1 abundance of each virus treated with DMSO was set to 1. All results are represented as means plus standard error from three independent infections (symbols represent biological replicates). Statistically significant differences (*$P < 0.05$) between the indicated condition and the corresponding data point of WA.1 or DMSO were determined by the student's unpaired two-sided *t* test (exact *p* values from left to right: **B** 0.0002, 0.0001, 0.0001; **C** 0.0434; **E** 0.0001, 0.0017, 0.0002) or one-way ANOVA adjusted for multiple comparisons (**D** 0.96, 0.004). Refer to Supporting Dataset 1 for non-normalized data. Source data are provided as a Source Data file.

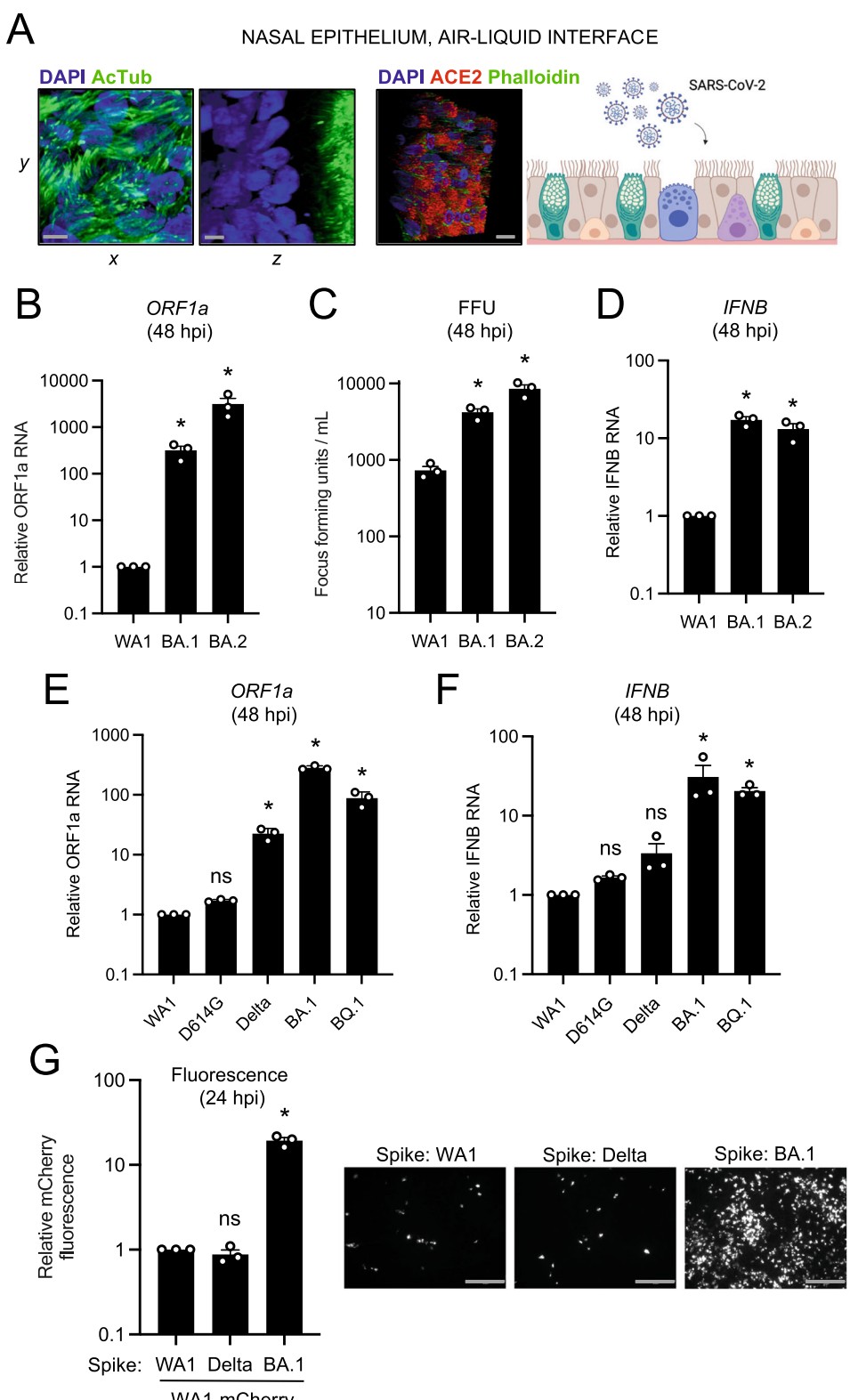

(Supplemental Fig. 3B), BA.1 and BQ.1 induced more *IFNB* expression compared to the other variants (Fig. 2F). We also inoculated nasal ALI with Omicron XBB, the ancestor of sublineages currently circulating in humans as of October 2023, and it replicated to a significantly greater extent than D614G (Supplemental Fig. 3C).

To determine whether the enhanced nasal tropism of Omicron BA.1 can be attributed to its Spike protein, we generated mCherry-expressing, recombinant SARS-CoV-2 WA1 encoding Spike from WA1,

Delta, or BA.1 (referred to as WA1 (WA.1 Spike), WA1 (Delta Spike), or WA1 (BA.1 Spike). We found that the presence of Spike from BA.1, but not Delta, resulted in a significant gain in nasal cell infectivity relative to WA1 (20-fold by mCherry fluorescence (Fig. 2G) or 100-fold by viral ORF1a RT-qPCR at 24 h post-inoculation (Supplemental Fig. 3D)). Results were similar regardless of how we calculated virus titers to prepare inoculants (infection of Vero E6-ACE2-TMPRSS2 cells (Supplemental Fig. 3D) or viral ORF1a RT-qPCR (Supplemental Fig. 3E).

**Fig. 2 | Omicron Spike enables enhanced SARS-CoV-2 infectivity in primary nasal epithelia cultured at the air–liquid interface. A** Primary human nasal epithelial cells (pooled from 14 donors) were cultured at the air–liquid interface and challenged with SARS-CoV-2. Cells were fixed and permeabilized for confocal immunofluorescence microscopy and stained with anti-AcTub, anti-ACE2, phalloidin, and DAPI. Stacked 3D confocal images in the *xy*, *yz*, or *xyz* fields are shown. Scale bars = 10 μm. Cartoon made with Biorender.com. **B** RNA was extracted from cells at 48 h post inoculation with 10,000 plaque-forming units of WA1, BA.1, or BA.2, and RT-qPCR of viral ORF1a was performed. Relative ORF1a transcript abundance compared to actin was determined by the $2^{(-\Delta\Delta CT)}$ method. ORF1a abundance of WA1 was set to 1. **C** Infectious virus from culture medium collected at 48 h post-inoculation was measured by challenging Vero E6-ACE2-TMPRSS2 followed by fixation/permeabilization, anti-N immunostaining, and high-content imaging. Symbols represent results from three independent infections of Vero E6-ACE2-TMPRSS2 cells. **D** RT-qPCR of cellular *IFNB* was performed at 48 h post-inoculation. *IFNB* abundance of WA1 was set to 1. **E** Viral ORF1a RT-qPCR was performed at 48 h post-inoculation with 10,000 plaque-forming units of WA1, D614G, Delta, BA.1, or BQ.1. ORF1a abundance of WA1 was set to 1. **F** RT-qPCR of cellular *IFNB* was performed at 48 h post-inoculation. *IFNB* abundance of WA1 was set to 1. **G** Recombinant WA1 (WA1 Spike), WA1 (Delta Spike), or WA1 (BA.1 Spike) were used to inoculate primary human nasal epithelial cells cultured at the air–liquid interface, and infection was measured by mCherry fluorescence at 24 h post-inoculation. The fluorescence intensity of WA1 (WA1 Spike) was set to 1. Representative fields of view are shown. Scale bars = 300 μm. All results are represented as means plus standard error from three independent infections (biological replicates). Statistically significant differences (*$P < 0.05$) between the indicated condition and the corresponding condition of WA.1 were determined by one-way ANOVA adjusted for multiple comparisons (exact *p* values from left to right: **B** 0.0206, 0.0323; **C** 0.0265, 0.0005; **D** 0.0009, 0.0041; **E** 0.999, 0.0017, 0.0001, 0.0003; **F** 0.999, 0.998, 0.0215, 0.007; **G** 0.996, 0.0001). Refer to Supporting Dataset 1 for non-normalized data. Source data are provided as a Source Data file.

When earlier timepoints were examined, WA1 (BA.1 Spike)-infected cells were observed as early as 6 hours post-inoculation and numbers increased substantially between 9 and 12 h post-inoculation, indicative of rapid virus spread in nasal ALI (Supplemental Fig. 4A, B). These results formally indicate that the elevated nasal tissue infectivity of BA.1 is governed by its Spike protein and, more specifically, by the mutations that distinguish it from Delta Spike. Therefore, enhanced nasal cell tropism evolved recently in the Omicron ancestor as a result of unique mutations in Spike.

## Entry of Omicron BA.1 into primary nasal epithelial cells is mediated by proteases of the MMP/ADAM families

Since BA.1 Spike encodes for an unprecedented ability to enter human nasal epithelia, we characterized the subcellular location at which Omicron Spike-mediated entry occurs in nasal epithelia and determined the cellular proteases that enable it. Several reports claim that Omicron enters cells via cathepsin-dependent fusion in endosomes[31,32,36], but the Omicron entry pathway may vary in transformed cell lines[44]. Therefore, we examined the entry determinants of WA1, Delta, and BA.1 in primary human nasal ALI. We used E64d to disrupt cathepsin-mediated entry in endolysosomes and found that, while WA1 replication was reduced by 25%, that of Delta and BA.1 were not (Fig. 3A). Interestingly, Delta and BA.1 infections were enhanced approximately 2-fold by E64d treatment, despite the same dose being previously shown to partially inhibit Omicron infection in a transformed lung cell line[31]. However, the addition of camostat mesylate (an inhibitor of serine transmembrane proteases including TMPRSS2) alone or in combination with E64d strongly blocked infection of WA1 and Delta in nasal ALI (Fig. 3A). These data suggest that WA1 and Delta enter nasal ALI primarily through a TMPRSS2-dependent entry route at the plasma membrane. In contrast, infection of nasal ALI by BA.1 was completely insensitive to blockade by camostat (Fig. 3A), suggesting the use of a divergent entry pathway that is TMPRSS2-independent. The TMPRSS2-independence of Omicron infection has been reported by multiple groups, and as a result, it has been inferred that Omicron uses an endosomal, cathepsin-dependent route to enter human cells[31,32,36]. However, a recently described metalloproteinase-mediated entry pathway for SARS-CoV-2 that may enable virus fusion at the plasma membrane[11,45] prompted us to test the effect of incyclinide, a broad-spectrum inhibitor of MMP and ADAM metalloproteinases. WA1, Delta, and BA.1 infections were reduced by incyclinide treatment in primary ALI, but BA.1 was particularly sensitive (inhibited by 27-fold compared to 4-fold inhibition of WA1 and 5-fold inhibition of Delta) (Fig. 3A). These findings suggest that processing of BA.1 Spike by metalloproteinases, but not by TMPRSS2, enables BA.1 entry into primary nasal epithelia. To address whether a metalloproteinase-dependent route of entry allows for fusion at the plasma membrane

or at endolysosomal membranes, we blocked endosomal trafficking with the endolysosomal acidification inhibitor bafilomycin A1. WA1 and Delta infections were inhibited by bafilomycin A1, suggesting that these viruses can utilize an endocytic pathway for entry (Fig. 3A). In contrast, bafilomycin A1 enhanced BA.1 infection 4-fold, suggesting that BA.1 may utilize an entry pathway that is non-endocytic (Fig. 3A). We also tested the inhibitor sensitivity of recombinant WA1 (Delta Spike) and WA1 (BA.1 Spike) to further clarify the entry pathway unlocked by BA.1 Spike in nasal ALI. As seen with the full-length clinical isolates, WA1 (Delta Spike) was inhibited by bafilomycin A1 while WA1 (BA.1 Spike) was not (Fig. 3B). However, WA1 (BA.1 Spike) was inhibited 10-fold by combined treatment of bafilomycin A1 and incyclinide. For WA1 (Delta Spike), treatment with bafilomycin A1 and incyclinide inhibited infection to the same extent as bafilomycin A1 alone (Fig. 3B). These results may suggest that metalloproteinases at the cell surface promote entry mediated by BA.1 Spike, while those in endosomes may promote entry mediated by Delta Spike. Overall, our data support a model whereby Delta Spike may be processed by TMPRSS2 at the plasma membrane or by metalloproteinases in endosomes, allowing for fusion at both sites in nasal ALI. In contrast, BA.1 Spike may be processed primarily by metalloproteinases at or near the cell surface, and fusion ensues in this compartment, not in endosomes.

To further substantiate a role for metalloproteinases in the entry process driven by different Spikes in nasal ALI, we tested the sensitivity of WA (Delta Spike) and WA1 (BA.1 Spike) to additional metalloproteinase inhibitors. Apratastat is an inhibitor of ADAM10, ADAM17 and MMP-13, and batimastat is an inhibitor of MMP-1, MMP-2, MMP-3, MMP-7, MMP-9, ADAM10, and ADAM17. While incyclinide inhibited WA1 (Delta Spike) infection, apratastat and batimastat did not inhibit but instead modestly boosted infection (Fig. 3C). In contrast, both apratastat and batimastat reduced WA (BA.1 Spike) infection by 60% by (Fig. 3C). These results may suggest that, compared to Delta Spike, BA.1 Spike is more dependent on processing by metalloproteinases. Therefore, BA.1 Spike has evolved to utilize a distinct entry route in nasal epithelia that is TMPRSS2-independent and metalloproteinase-dependent. To assess whether preferential usage of neuropilin-1 at the cell surface by BA.1 Spike may explain the increased adherence of BA.1 to nasal cell epithelia and adoption of a unique entry pathway, we tested the impact of the neuropilin-1 inhibitor EG00229 on nasal ALI infection by WA1 (WA1 Spike), WA1 (Delta Spike), and WA1 (BA.1 Spike). WA1-Spike was most sensitive to inhibition by EG00229, while Delta Spike was completely resistant, and BA.1 Spike conferred modest sensitivity (Supplemental Fig. 5). These results indicate that the extent of neuropilin-1 dependency does not explain the differential ability for WA1 Spike, Delta Spike, and BA.1 Spike to mediate entry into nasal epithelia. Therefore, factors other than neuropilin-1 may be responsible for the improved infectivity of BA.1 in nasal tissue.

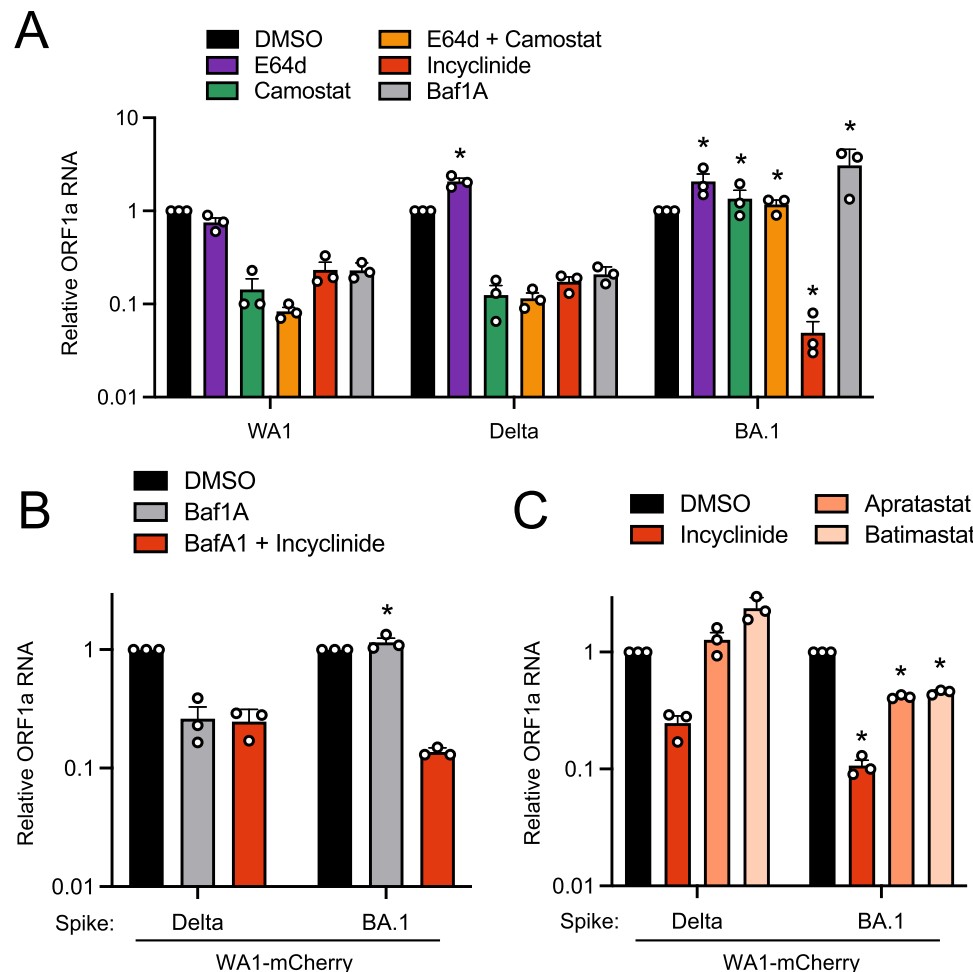

**Fig. 3 | Entry of Omicron BA.1 into primary nasal epithelial cells is sensitive to MMP inhibitors but not inhibitors of cathepsins or TMPRSS2. A** Primary human nasal epithelial cells (pooled from 14 donors) were cultured at the air-liquid interface and pretreated with 10 μM E64d, 10 μM camostat mesylate, a combination of 10 μM E64d and 10 μM camostat mesylate, 10 μM incyclinide, or 1 μM bafilomycin A1 for 2 h. Cells were challenged with WA.1, Delta, or BA.1 at an MOI of 0.05. At 48 h post-inoculation, RNA was extracted from cells, and ORF1a levels were measured by RT-qPCR. For each virus, ORF1a abundance in the DMSO condition was set to 1. Results are represented as means plus standard error from three independent infections (biological replicates). Statistically significant differences (*$P < 0.05$) between the indicated condition and the corresponding condition of WA.1 were determined by one-way ANOVA adjusted for multiple comparisons (exact $p$ values from left to right: 0.0025, 0.0372, 0.019, 0.0013, 0.0236, 0.0313). **B** Primary human nasal epithelial cells were cultured at the air-liquid interface and pretreated with 1 μM bafilomycin A1 or 1 μM bafilomycin A1 and 10 μM incyclinide for 2 h. Cells were challenged with WA1 (Delta Spike) or WA1 (BA.1 Spike) at an MOI of 0.05. At 48 h post-inoculation, ORF1a levels were measured by RT-qPCR. For each virus, ORF1a abundance in the DMSO condition was set to 1. Statistically significant differences (*$P < 0.05$) between the indicated condition and the corresponding condition of WA.1 (Delta Spike) were determined by the student's unpaired two-sided $t$ test (exact $p$ value: 0.0015). **C** As in **B**, except primary human nasal epithelial cells cultured at the air-liquid interface were pretreated with 10 μM incyclinide, 10 μM apratastat, or 10 μM batimastat for 2 h and challenged with WA1 (Delta Spike) or WA1 (BA.1 Spike). Statistically significant differences (*$P < 0.05$) between the indicated condition and the corresponding condition of WA1 (Delta Spike) were determined by the student's unpaired two-sided $t$ test (exact $p$ values from left to right: 0.0254, 0.0121, 0.0038). All results are represented as means plus standard error from three independent infections (symbols represent biological replicates). Refer to Supporting Dataset 1 for non-normalized data. Source data are provided as a Source Data file.

## Omicron Spike enables an entry pathway into primary nasal epithelial cells that is resistant to inhibition by type-I and type-III interferons

Next, we explored whether the increased nasal cell infectivity exhibited by Omicron is related to evasion of or resistance to cell-intrinsic antiviral defenses. During the preparation of this manuscript, it was demonstrated that SARS-CoV-2 variants of concern displayed an increasing resistance to inhibition by interferon (IFN) treatment, with Omicron exhibiting the greatest degree of resistance[46]. However, the role of Spike in conferring IFN resistance was not assessed, and it was unknown whether an Omicron-specific cellular entry pathway enabled IFN resistance in nasal epithelia. Therefore, we tested whether the unique entry requirements of Omicron Spike allow for evasion of the IFN-induced antiviral state in primary nasal ALI (Fig. 4A). Pre-treatment with IFN-beta or IFN-lambda inhibited WA1 in a dose-dependent

manner (up to two orders of magnitude) as measured by viral ORF1a RT-qPCR, while BA.1 or BA.2 infections were significantly less impacted (Fig. 4B). We also quantified infectious virus yield under these conditions and confirmed that BA.1 and BA.2 achieved higher titers and were inhibited by IFN to a significantly lesser extent than WA1 (Fig. 4C). When we compared the sensitivity of full-length clinical virus isolates with recombinant WA1 encoding different Spike proteins, we found that the identity of Spike governed sensitivity to IFN. Compared to WA1, which was inhibited sixfold by low amounts of IFN-beta or IFN-lambda, WA1 (Delta Spike) exhibited a slight decrease in sensitivity to IFN (Fig. 4D). However, WA1 (BA.1 Spike) was completely insensitive to the same doses of IFN, and this closely resembled the resistance profile of full-length BA.1 (Fig. 4D). Therefore, BA.1 Spike is sufficient to confer IFN-resistance to an otherwise IFN-sensitive virus, while Delta Spike is not. Moreover, these results reveal the presence of host factors

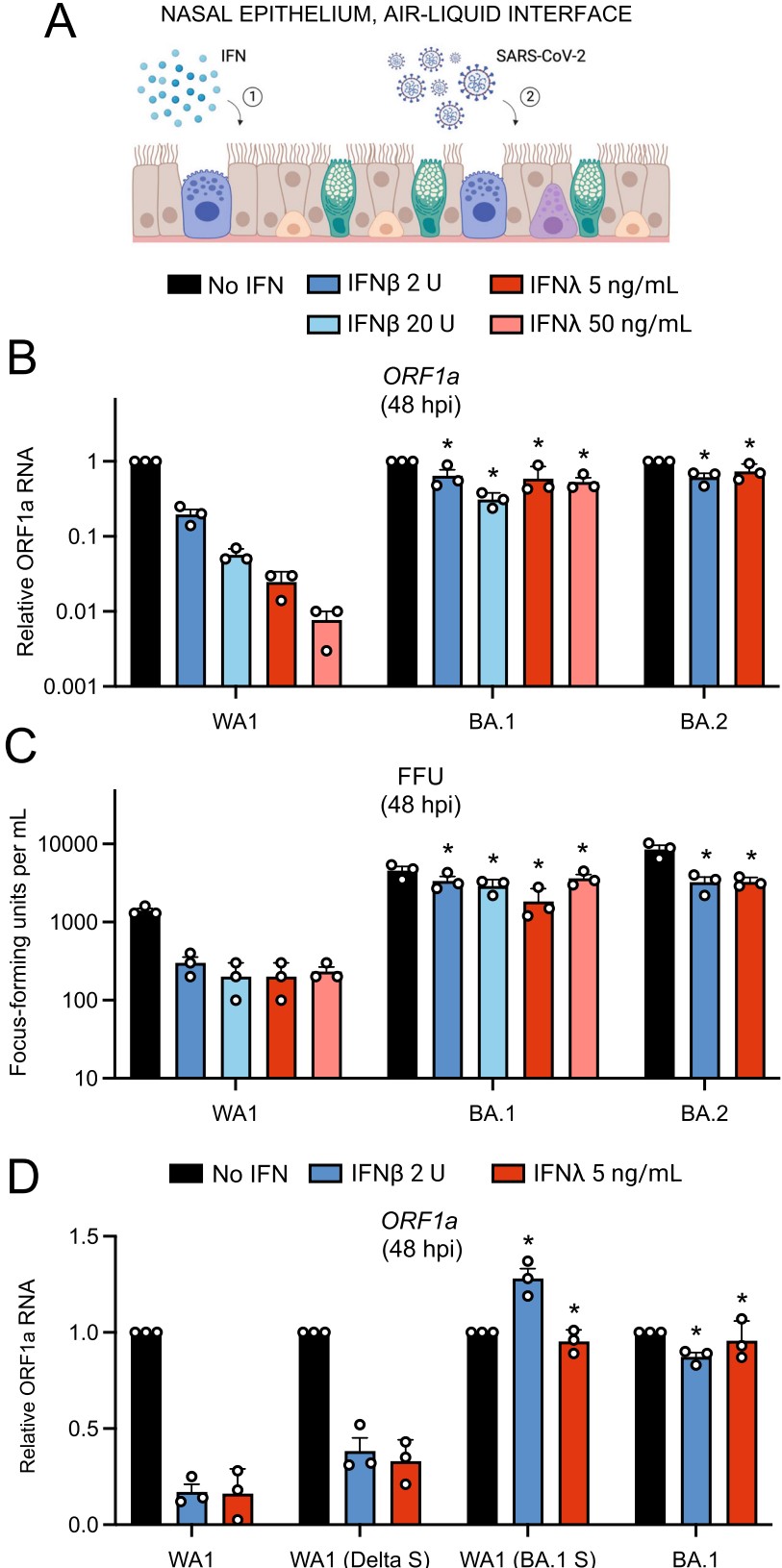

intrinsic to nasal cells that are interferon-inducible and target the Spike-mediated entry step.

We also examined the IFN sensitivity of WA1, BA.1, and BA.2 in submerged primary nasal monolayers (Supplemental Fig. 6A). While IFN-beta or IFN-lambda exposure strongly blocked WA1 infection by more than 100-fold, the same amount of IFN inhibited BA.1 and BA.2 to

a much lesser extent (less than 2-fold) (Supplemental Fig. 6B). To confirm that Spike is a major determinant of IFN sensitivity in these cells, we generated pseudovirus consisting of vesicular stomatitis virus (VSV) decorated with Spike from WA1, BA.1, or BA.2. VSV-WA1 Spike was exquisitely sensitive to inhibition by IFN-beta or IFN-lambda in a dose-dependent manner, with inhibition approaching three orders of

**Fig. 4 | Omicron Spike enables an entry pathway into primary nasal epithelial cells that is resistant to inhibition by type-I and type-III interferons. A** Primary human nasal epithelial cells (pooled from 14 donors) were cultured at the air-liquid interface, pre-treated for 18 h with the indicated amounts of IFN-beta or IFN-lambda, and challenged with 10,000 plaque-forming units of WA1, BA.1, or BA.2. Cartoon made with Biorender.com. **B** At 48 h post inoculation, total RNA was extracted from cells and subjected to RT-qPCR. ORF1a levels were measured compared to actin by the $2^{(-\Delta\Delta CT)}$ method. For each virus, ORF1a abundance in the No IFN condition was set to 1. **C** At 48 h post inoculation, infectious virus titers from the cell culture medium were measured by challenging Vero E6-ACE2-TMPRSS2 cells followed by fixation/permeabilization and immunostaining with anti-N antibody. Infectious units were quantified by measuring focus forming units by high-content imaging. Symbols represent independent infections of Vero E6-ACE2-TMPRSS2 cells. **D** Primary human nasal epithelial cells (pooled from 14 donors) were cultured at the air–liquid interface, pre-treated for 18 h with the indicated amounts of IFN-beta or IFN-lambda, and challenged with 10,000 plaque-forming units of WA1, WA1 (Delta Spike), WA1 (BA.1 Spike), or BA.1. ORF1a levels were measured by RT-qPCR and compared to actin by the $2^{(-\Delta\Delta CT)}$ method. For each virus, ORF1a abundance in the No IFN condition was set to 1. All results are represented as means plus standard error from three independent infections (symbols represent biological replicates). Statistically significant differences (*$P < 0.05$) between the indicated condition and the corresponding data point of WA.1 were determined by the student's unpaired two-sided $t$ test (exact $p$ values from left to right: **B** 0.0288, 0.00388, 0.0197, 0.0022, 0.0063, 0.0028; **C** 0.0032, 0.0016, 0.0298, 0.0016, 0.0056, 0.0040; **D** 0.0001, 0.0001, 0.0001, 0.0001). Refer to Supporting Dataset 1 for non-normalized data. Source data are provided as a Source Data file.

magnitude (Supplemental Fig. 6C). In contrast, VSV-BA.1 Spike and VSV-BA.2 Spike were much less sensitive to inhibition by IFN-beta or IFN-lambda (Supplemental Fig. 6C). Therefore, Spike is the major determinant of the IFN resistance displayed by Omicron sublineages, and this feature can be transferred to other viruses by pseudotyping. Additionally, we determined that cellular attachment of WA1 and BA.1 in nasal monolayers was not affected by IFN-beta or IFN-lambda, indicating that nasal cell-intrinsic host defenses act at a post-attachment step of the virus entry process (Supplemental Fig. 6D). Therefore, Omicron Spike enables evasion of restriction by nasal cell-intrinsic host factors inhibiting the post-attachment step of virus entry.

### Amphotericin B partially relieves a block to infection mediated by WA1 Spike and Delta Spike in primary nasal epithelia

A limited number of interferon-stimulated genes are known to block the cellular entry step of enveloped viruses[47]. Among the best characterized are the interferon-induced transmembrane (IFITM) proteins. IFITM proteins provide constitutive and interferon-induced antiviral protection in many cell types by inhibiting the virus-cell membrane fusion step required for entry[47,48]. While it is established that murine IFITM3 restricts SARS-CoV-2 infection and limits viral disease in vivo[49,50], the roles played by human IFITM proteins during SARS-CoV-2 infection have remained controversial. For example, it was demonstrated that IFITM proteins can inhibit SARS-CoV-2 infection in endosomes but can promote infection at the cell surface[51,52]. Furthermore, it has been suggested that endogenous human IFITM proteins in lung epithelial cells enable SARS-CoV-2 infection by possibly acting as a receptor or attachment factor[53]. To address whether IFITM proteins act as a barrier to SARS-CoV-2 infection in primary human nasal epithelia, we tested the impact of the antifungal Amphotericin B on infection with recombinant WA1 (WA1 Spike), WA1 (Delta Spike), and WA1 (BA.1 Spike). Amphotericin B was previously reported to counteract the antiviral properties of human IFITM2 and IFITM3[54–56] by restoring membrane fluidity in cellular membranes where IFITM3 is present[57]. We found that nasal ALI constitutively express ample amounts of IFITM2/3 protein (Fig. 5A). In nasal ALI pretreated with Amphotericin B or left untreated, we inoculated with WA1 (WA1 Spike), WA1 (Delta Spike), or WA1 (BA.1 Spike) and scored infection by mCherry fluorescence at 24 hours post-inoculation. As described above, WA1 (BA.1 Spike) displayed a far superior capacity to infect untreated nasal ALI relative to WA1 (WA1 Spike) and WA1 (Delta Spike) (Fig. 5B, C). However, Amphotericin B treatment led to marked increases in WA1 (WA1 Spike) and WA1 (Delta Spike) infections, boosting them by approximately 7-fold. By comparison, Amphotericin B had a negligible impact on WA1 (BA.1 Spike) infection at the same time point (Fig. 5B, C). We also determined whether Amphotericin B impacted WA1 (BA.1 Spike) infection at earlier time points when infected cells were less abundant. At 9 and 12 h post-inoculation, Amphotericin B marginally boosted infection of WA1 (BA.1 Spike) (less than 2-fold) (Supplemental Fig. 7A, B). Our measurements of mCherry fluorescence

produced by WA1 (BA.1 Spike) were not compromised by fluorescence saturation because saturation was only evident at 36 h post-inoculation (Supplemental Fig. 7C). Therefore, these results suggest that human IFITM proteins restrict WA1 (WA1 Spike) and WA1 (Delta Spike) in primary nasal epithelia, while BA.1 Spike confers escape from this restriction. Therefore, antiviral IFITM proteins form a part of the nasal cell-intrinsic defenses that limit infection by early/ancestral SARS-CoV-2.

Overall, by combining experiments with authentic SARS-CoV-2, recombinant virus, and Spike-decorated pseudovirus, we show that Spike is the major determinant of the enhanced infectivity and interferon resistance exhibited by Omicron in primary human nasal epithelia, two phenotypes which are associated with a unique dependence on metalloproteinases for cell entry at or near the cell surface.

## Discussion

Using entirely primary nasal epithelia from human adult donors, we demonstrate that Omicron (including BA.1, BA.2, BQ.1, and XBB) exhibits drastically increased infectivity in nasal tissue compared to early SARS-CoV-2, the D614G variant, and the Delta variant of concern. This mapped to Omicron Spike and, at least in part, to an increased ability for Omicron virions to adhere to the surface of nasal epithelia, which occurred independently of the presence of cilia. Furthermore, we show that Omicron utilizes an entry route into cells that depends on cellular metalloproteinases, but that is independent of the transmembrane serine protease TMPRSS2. Since we also demonstrate that Omicron Spike-mediated infection enables evasion of the antiviral state induced by type-I and type-III IFN in this tissue, these results suggest that the metalloproteinase-dependent entry pathway utilized by Omicron may promote escape from constitutive and IFN-induced antiviral factors targeting virus entry.

During the preparation of this manuscript, it was reported that BA.1 exhibits a growth advantage in primary human nasal epithelia compared to the Delta variant[31]. Here, using recombinant SARS-CoV-2, we extend this finding by demonstrating that BA.1 Spike is responsible for increased nasal cell infectivity. However, in another study, while BA.1 exhibited a gain in nasal cell infectivity compared to Alpha; no significant differences in nasal cell infectivity were observed between Delta and BA.1[32]. This discrepancy is likely the result of the utilization of distinct cell culture models to compare relative infectivity. Our use of recombinant, fluorogenic SARS-CoV-2 in nasal ALI derived from pools of 14 human adult donors shows that Spike from BA.1 promotes a gain in nasal cell infectivity compared to Delta Spike. Therefore, the emergence and evolution of Omicron involved unique mutations in Spike that contributed to enhanced nasal cell binding and entry, and this may explain how Omicron replaced previously circulating variants of concern and continues to persist in human populations.

It was previously reported that human nasal airways express relatively low levels of TMPRSS2, and combined with the demonstration that Omicron exhibits TMPRSS2-independence in transformed cell lines, others have inferred that Omicron is more reliant on

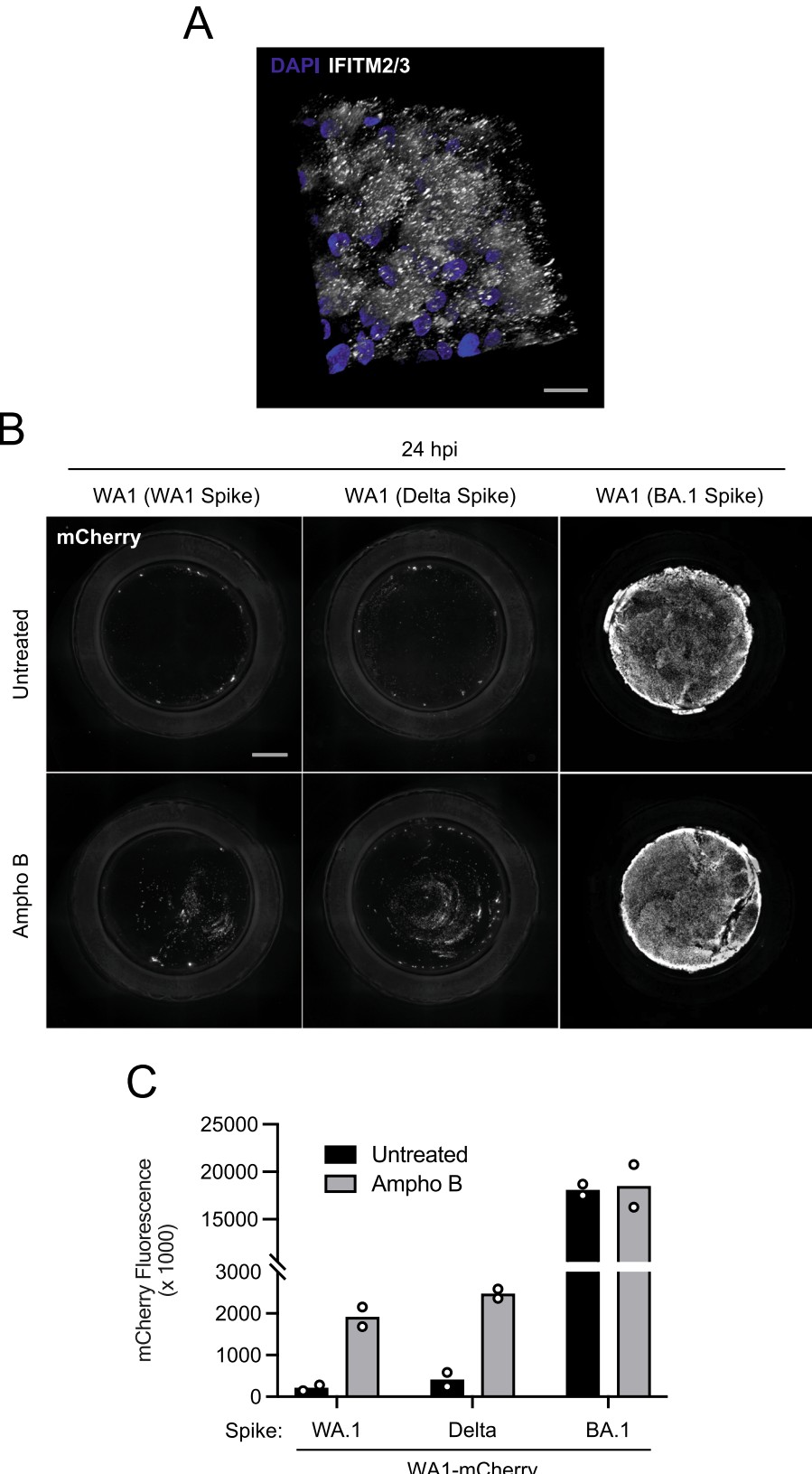

**Fig. 5 | Amphotericin B partially relieves a block to WA1 Spike- and Delta Spike-mediated infection in primary nasal epithelia. A** Primary human nasal epithelial cells (pooled from 14 donors) were cultured at the air–liquid interface. IFITM2/3 levels were determined by anti-IFITM2/3 immunofluorescence, and DAPI was used to stain nuclei. A 3D reconstruction (*xyz*) of stacked confocal images is shown. **B** Primary human nasal epithelial cells (pooled from 14 donors) were cultured at the air–liquid interface, pre-treated with 1 μM Amphotericin B for 2 h or untreated, and inoculated with 10,000 plaque-forming units of WA1 (WA1 Spike), WA1 (Delta Spike), or WA1 (BA.1 Spike). mCherry fluorescence was measured in fixed tissue at 24 h post-inoculation by high-content imaging of the entire well. **C** Quantification of mCherry fluorescence intensity from two independent experiments is shown. Scale bar = 300 μm. Ampho B; amphotericin B. Source data are provided as a Source Data file.

endosomal cathepsins for entry[32]. However, our findings with primary nasal epithelia reveal that TMPRSS2-independence does not necessarily imply the use of an endosomal entry route by Omicron. Metalloproteinases such as MMP-14, MMP-16, ADAM10, and ADAM17 are also highly expressed in upper airways and have been shown to activate Omicron Spike-mediated fusion at the surface of several transformed cell lines[11]. Here, we demonstrate that the primary route of entry of Omicron in primary nasal epithelia is mediated by metalloproteinases. The pan-metalloproteinase inhibitor incyclinide strongly blocked BA.1 infection in nasal ALI, but the endosomal cathepsin inhibitor E64d and the endolysosomal acidification inhibitor bafilomycin A1 did not. Instead, these endosomal inhibitors boosted infection of BA.1 in nasal epithelia, suggesting that the endosomal entry route is deleterious for BA.1 in this tissue. Bafilomycin A1 inhibits vacuolar-ATPase to prevent endolysosomal acidification, and furthermore, endosomal maturation and transport between early and late endosomes is repressed[58]. Therefore, our results suggest that processing of Spike by metalloproteinases enables fusion at the plasma membrane and/or in early or recycling endosomes, while late endosomes or lysosomes and the cathepsins found therein are unlikely to be involved.

One possibility explaining the enhanced capacity for Omicron Spike to adhere to and enter nasal cells (either in the presence or absence of ciliated cells) is its increase in net overall positive charge compared to previous variants of concern[59]. It has also been proposed that Omicron Spike may adhere more strongly to attachment factors present at the epithelial cell surface, including heparan sulfate[60,61] and neuropilin-1[62]. In this study, sensitivity to neuropilin-1 inhibitor did not correlate with the nasal cell infectivity of WA1, Delta, or BA.1, suggesting that other cellular factors may play a role (Supplemental Fig. 3). A significant aim of future research will be to establish whether mutations in Omicron Spike promote increased residence time at the cell surface and whether this influences its dependence on certain cellular proteases for cleavage and the triggering of fusion. It is possible that longer residency at the cell surface may negatively interfere with processing by TMPRSS2 and steer processing towards metalloproteinases.

Another factor that likely contributed to the spread and dominance of Omicron lineages is their resistance to the antiviral properties of IFN. Early/ancestral SARS-CoV-2 was previously reported to antagonize IFN signaling in cells, and nearly every viral protein encoded by SARS-CoV-2 has been shown to contribute to the evasion of IFN through various mechanisms[63–65]. While the relative insensitivity of Omicron to exogenous administration of multiple IFN types has been reported previously[46,66], the viral proteins responsible for rendering IFN ineffective against Omicron were unknown. Other studies showed that Omicron exhibits a decreased capacity to interfere with IFN production and signaling relative to early/ancestral SARS-CoV-2 or Delta[44,67–69], and our measurements of type-I IFN production by Omicron-infected nasal ALI affirm this finding. However, in one case, Omicron was purported to induce a relatively inferior IFN response compared to Delta[66], and this discrepancy underlines the importance of assessing this activity in primary human upper airway epithelia. Overall, we postulate that Omicron exhibits resistance to IFN not because of its ability to interfere with the production or function of IFN from infected cells but rather because Omicron Spike enables passage into nasal cells in a manner that is insensitive to IFN-stimulated proteins. Furthermore, our findings provide a new explanation for why Omicron exhibits decreased IFN antagonism compared to previous variants of concern—if the Spike-mediated entry process of Omicron is resistant to type-I and type-III IFN, the selective advantage to antagonize IFN pathways may be lost. Furthermore, the improved capacity for Omicron lineages to infect nasal cells despite the induction of an antiviral state therein provides a plausible scenario for how Omicron displaced previous variants of concern. It was demonstrated that IFN produced from Omicron-infected cells can inhibit infection by Influenza A virus[67], so it is plausible that the IFN response elicited by Omicron infection may render nasal epithelia resistant to infection by previous SARS-CoV-2 variants of concern.

It is also possible that Omicron displays enhanced infectivity in nasal cells because Omicron Spike confers resistance to antiviral factors that are constitutively expressed in nasal epithelia and further upregulated by type-I or type-III IFN. Our results indicate that Amphotericin B partially relieves a constitutive barrier to infection mediated by IFITM proteins and provides direct evidence for this possibility. It is worth noting that our results showing that BA.1 Spike-mediated infection is resistant to the antiviral activities of IFITM proteins in nasal epithelia are not in agreement with recent publications[70,71]. It is possible that the impacts of IFITM proteins on SARS-CoV-2 infection may vary depending on cell type (transformed versus primary), tissue type (lung versus nasal), and virus type (pseudotypes versus replication-competent clinical isolates).

Our description of the IFN resistance conferred by Omicron Spike has implications for the ongoing development of type-I and type-III IFN as antiviral therapeutics for the prevention or treatment of SARS-CoV-2 infection and COVID-19. A case has been made for the deployment of type-III IFN in the context of SARS-CoV-2 infection since the receptor for type-III IFN is expressed in the respiratory tract but is not widely distributed elsewhere (resulting in fewer unintended side effects such as inflammation)[63,72]. As an important proof-of-concept, nasal administration of IFN-lambda was protective against SARS-CoV-2 variants, including Omicron, in mice[73]. However, clinical trials in humans have reported conflicting findings thus far. One such study reported that participants who were administered a single injection of IFN-lambda following symptom onset showed no significant reductions in viral shedding or symptom severity compared to placebo[74], while another reported that a single injection upon symptom onset resulted in reduced viral loads[75] and reduced COVID-19-related hospitalizations[76]. However, in neither case was it determined how IFN-lambda fared in individuals infected with Omicron variants. Since Omicron displays a greater degree of IFN resistance compared to earlier variants, additional clinical studies are needed to test the suitability of IFN-lambda in the fight against contemporaneous Omicron and additional variants that arise in the future.

While some mutations in Omicron Spike may have conferred Omicron with superior evasion of vaccine-elicited neutralizing antibodies compared to the Delta variant, our findings indicate that mutations in Omicron Spike also allow for efficient seeding of infection in the nasal epithelium and resistance to the interferon-induced antiviral state therein. Our findings provide mechanistic insight into the basis for the efficient transmission and persistence of Omicron in vaccinated and unvaccinated human populations. Future efforts aimed at understanding how individual Spike mutations contribute to these different phenotypes may enable a better understanding of the selective forces driving their emergence.

## Methods
### Tissue culture
Frozen primary human nasal airway epithelial cells (hNAEC) were purchased from Epithelix and cultured as submerged monolayers with hAEC Culture Medium (Epithelix). hNAEC from three human donors were thawed and pooled, and cells were passed every 5 days. Fresh primary human nasal airway epithelial cells cultured at the liquid-air interface (nasal ALI) were obtained from Epithelix (MucilAir, Pool of Donors) and cultured with MucilAir culture medium (Epithelix) in trans-well plates provided by the company; culture medium was replaced every 2 days. Nasal ALI cultures were maintained for 2–5 weeks in our laboratories prior to inoculation with SARS-CoV-2. hNAEC and Nasal ALI were fixed with 4% paraformaldehyde, permeabilized with 0.1% Triton-X, and stained with anti-ACE2 (Abcam; ab15348), anti-Acetyl-a-Tubulin (Lys40) (D20G3) (Cell Signaling; 5335),

anti-IFITM2/3 (Proteintech; 66081-1-Ig), phalloidin (Cell Signaling; 8878), and DAPI to assess ACE2 levels, the presence of ciliated cells, IFITM2/3 levels, and pseudostratification of nuclei, respectively. DyLight 488 (Invitrogen; 35502) or DyLight 680 (Invitrogen; 35519) were used as secondary antibodies. Images were obtained using the Zeiss LSM 880 workstation and AiryScan capability and were processed into 3D reconstructions by volume rendering using Imaris software. Primary human small airway (lung) epithelial cells (hSAEC) were purchased from ATCC (PCS-301-010) and were cultured with Airway Epithelial Cell Basal Medium (ATCC, PCS-300-030) and the Bronchial Epithelial Growth Kit (ATCC, PCS-300-400). hSAEC from three human donors were thawed and pooled, and cells were passed every 5 days. Vero E6-ACE2-TMPRSS2 cells were obtained from BEI Resources (NR-54970). HEK293T-ACE2 cells were obtained from BEI Resources (NR-52511).

## Production of SARS-CoV-2 variants and infections

SARS-CoV-2 isolate USA/WA1/2020 (WA1; NR-52281), isolate hCoV-19/ USA/HI-CDC-4359259-001/2021 (Lineage B.1.1.529; Omicron BA.1 variant; NR-56475), and isolate hCoV-19/USA/NY-MSHSPSP-PV56475/2022 (Lineage BA.2.12.1; Omicron BA.2 variant; NR-56782), isolate hCoV-19/ USA/CA-Stanford-109_S21/2022 (Lineage XBB; Omicron XBB variant; NR-58927) and hCoV-19/USA/MDHP38960/2022 (Lineage BQ; Omicron BQ.1 variant) were obtained from BEI Resources. hCoV-19/New York-PV08410/2020 (Lineage B.1; D614G variant; NR-53514) was a gift from John Dye (USAMRIID). Isolate hCoV-19/USA/MD-HP05285/2021 VOC Delta G/478 K.V1 (Lineage B.1.617.2 + AY.1 + AY.2; Delta variant) was obtained from Andrew Pekosz (Johns Hopkins University). Recombinant WA1-mCherry virus was rescued from a SARS-CoV-2 cDNA construct, and mCherry was inserted at the N-terminus of the *N* gene. A P2A linker was placed between mCherry and N. To generate WA1-mCherry (BA.1 Spike) and WA1-mCherry (Delta Spike), the sequence encoding WA1 Spike was replaced with Spike from BA.1 or Delta, respectively. All viruses were grown in Vero E6-ACE2-TMPRSS2 cells, and harvested cell culture supernatants were titrated in Vero E6-ACE2-TMPRSS2 cells to calculate infectious titers. Infectious titers were determined by focus-forming unit assay whereby foci of infection were measured using an anti-nucleocapsid (N) antibody (Invitrogen; MA1-7403). Where indicated, ORF1a RNA titers were also calculated for some viruses to approximate virus particle quantity.

Infections of hNAEC or hSAEC monolayers were performed at a multiplicity of infection (MOI) of 0.05 as follows: cells were inoculated with virus suspension for 2 h at 37 °C; inoculum was removed; cells were washed with PBS and returned to 37 °C for the time indicated. For detection of virus attachment, hNAEC were inoculated with virus suspension for 1 h on ice; inoculum was removed; cells were washed three times with cold PBS.

For detection of infection or virus attachment by RT-qPCR, cells were lysed with Trizol (Sigma). Viral replication was measured using RT-qPCR amplification of viral ORF1a, as previously described[50]. Cells lysed with Trizol were mixed with chloroform (Sigma) at a 5:1 (Trizol:chloroform) ratio. Mixed samples were mixed thoroughly and incubated at room temperature for 10 min, followed by centrifugation at $12,000 \times g$ for 5 min to allow separation of the aqueous and organic phases. Equal volumes of 70% ethanol were added to the aqueous phases, mixed thoroughly, and incubated at room temperature for 5 min. RNA purification was performed using the PureLink RNA Mini Kit (Invitrogen) according to the manufacturer's instructions. Purified RNA product was immediately used with the One-step PrimeScript RT-PCR Kit (Takara). Primers and probes were obtained from IDT. The primers and probes used to amplify and quantify ORF1a are as follows (5'-3'): ORF1a-F AGAAGATTGGTTAGATGATGATAGT; ORF1a-R TTCC ATCTCTAATTGAGGTTGAACC; ORF1a-P FAM/TCCTCACTGCCGT CTTGTTGACCA/BHQ13. The primers and probes used to amplify and quantify beta-actin (ACTB) are as follows (5'-3'): ACTB-F ACAGA

GCCTCGCCTTTG; ACTB-R CCTTGCACATGCCGGAG; ACTB-P 56- FAM/ TCATCCATG/ZEN/GTGAGCTGGCGG/31ABkFQ. The primers and probes used to amplify and quantify IFITM3 are as follows (5'-3'): IFITM3-F ACCATGAATCACACTGTCCAAACCTT; IFITM3-R CCAGCACA GCCACCTCG; IFITM3-P FAM/ZEN-CTCTCCTGTCAACAGTGGCCAGC CCC-IBFQ. The primers and probes used to amplify and quantify IFNB are as follows (5'-3'): IFNB-F GAAACTGAAGATCTCCTAGCCT; IFNB-R GCCATCAGTCACTTAAACAGC; IFNB-P 56-FAM/TGAAGCAAT/ZEN/ TGTCCAGTCCCCAGAGG/3IABkFQ. Reaction mixtures of 20 μL (including 2.2 μL total RNA, 0.2 μM forward and reverse primers, and 0.1 μM probe) were subjected to reverse transcription (5 min at 45 °C, followed by 10 s at 95 °C) and 40 cycles of PCR (5 s at 95 °C followed by 20 s at 60 °C) in a CFX Opus 96 Real-Time PCR System (BioRad). Results were analyzed by the Comparative Ct Method (ΔΔCt Method). RNA levels for viral ORF1a, IFNB, and IFITM3 were normalized to cellular ACTB.

For infections of primary nasal epithelial cells cultured at the air-liquid interface, the apical (air-exposed) surface was gently rinsed with 100 μL cell culture medium to partially remove mucous layers. Virus suspension in a volume of 50 μL was added to the apical surface, and cells were incubated for 2 h at 37 °C. The inoculum was then removed, and 100 μL PBS was used to gently wash the apical surface. Cells were then returned to 37 °C. At 48 h post inoculation, cells were lysed with Trizol and subjected to RNA extraction and RT-qPCR as indicated above. To quantify the infectious virus particles produced into the cell culture supernatant, the medium was removed, 200 μL PBS was added to the apical surface, and cells were incubated at room temperature for 20 min before the PBS was recovered. A focus-forming units assay was performed by inoculating Vero E6-ACE2-TMPRSS2 cells with the recovered volume. At 7 h post inoculation, Vero E6-ACE2-TMPRSS2 cells were fixed with 4% paraformaldehyde and stained with anti-N (Invitrogen; MA1-7403). The number of fluorescent foci was measured using a Cytation 5 Cell Imaging Multimode Reader (BioTeK).

For detection of recombinant WA-mCherry infection, the Cytation 5 Cell Imaging Multimode Reader (BioTeK) was used to measure mCherry fluorescence at the indicated hours post inoculation. Quantification of mCherry fluorescence was performed by measuring integrated signal intensities in Fiji.

## Production of VSV-Spike pseudoviruses and infections

Spike sequences from WA.1, BA.1, or BA.2 were codon-optimized for expression in human cells, synthesized with a 6xHis tag on the amino terminus, and cloned into pcDNA3.1 (+) by GenScript. HEK293T cells were seeded in a 10 cm dish and transfected with 12 μg pcDNA3.1 Spike plasmid using Lipofectamine 2000 (Thermo Fisher). Forty-eight hours after transfection, the culture medium was removed from cells, and 1 mL VSV-luc/GFP plus VSV-G (seed particles) was added. Twenty-four hours after infection, virus supernatants were collected, clarified by centrifugation at $500\,g$ for 5 min, followed by filtration with a 45 μm filter, and stored at −80 °C. A total of 50 μL virus supernatants was added to submerged hNAEC from three pooled human donors, and 24 hours post inoculation, cells were lysed with Passive Lysis Buffer (Promega). Luciferase activity was measured on a Perkin Elmer MicroBeta 2450 microplate luminometer using the Luciferase Assay System (Promega). Fifty microlitre volumes of VSV-WA1, VSV-BA.1, and VSV-BA.2 were found to infect Vero E6-ACE2-TMPRSS2 cells to similar extents, suggestive of similar infectious titers.

## Interferons and inhibitors

Recombinant human IFN-β (beta) (300-02BC) and human IFN-λ (lambda) (300-02 L) were obtained from PeproTech and were used to test IFN sensitivity of full-length, authentic SARS-CoV-2 variants (IFN was added 18 h prior to inoculation and removed prior to virus addition). Recombinant human IFN-β (beta) 1a (11415-1) and human IFN-λ (lambda) (11725-1) were obtained from PBL Assay Science and were

used to test IFN sensitivity of VSV-based pseudoviruses (IFN was added 18 h prior to inoculation and removed prior to virus addition). The following inhibitors were reconstituted in DMSO: E64d (Sigma; E8640), camostat mesylate (Sigma; SML0057), incyclinide (MedChemExpress; HY-13648), apratastat (MedChemExpress; HY-119307), batimastat (SelleckChem: S7155), EG00229 (SelleckChem; E1119). Bafilomycin A1 (Sigma; SML1661) was received as a ready-made solution in DMSO, and amphotericin B (Sigma; A2942) was received as a ready-made solution in deionized water.

## Statistical analysis
Summary statistics and statistical tests were made using Graphpad Prism version 10.1.0.

## Biosafety statement
This study was conducted in compliance with all relevant local, state, and federal regulations. Approval for the generation and use of recombinant SARS-CoV-2 WA1-mCherry and its variants using Biosafety Level 3 practices was provided by the NIH Institutional Biosafety Committee following evaluation by the Dual Use Research of Concern Institutional Review Entity (case number RD-22-X1-11; PI: Jonathan W. Yewdell). All work involving infectious SARS-CoV-2 variants, including recombinant virus, was performed in Biosafety Level 3 facilities. All personnel working with viruses were trained with relevant safety and procedure-specific protocols, and their competency for performing the work was certified.

## Reporting summary
Further information on research design is available in the Nature Portfolio Reporting Summary linked to this article.

## Data availability
Datasets generated and/or analyzed during the current study are available in the paper or are appended as supplementary data. Source data are provided in this paper.

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

## Acknowledgements

We would like to thank Brett Eaton, Elena Postnikova, Michael Murphy, Sushma Bhosle, Julie Tran, and Jillian Geiger (NIAID, Integrated Research Facility) for reagents and protocols to quantify viral ORF1a by RT-qPCR, Vincent Munster (NIAID) and Michael Letko (Washington State University) for reagents and protocols to produce VSV-Spike pseudo-viruses, and Kim Peifley (NCI) for assistance in confocal immunofluorescence microscopy. Work in the laboratory of AAC was funded by the Intramural Research Program, NIH, NCI, Center for Cancer Research, and an Intramural Targeted Anti-COVID-19 Award from NIAID. Work in the laboratory of JWY was funded by the Division of Intramural Research, NIH, NIAID. The content of this publication does not necessarily reflect the views or policies of the Department of Health and Human Services, nor does mention of trade names, commercial products, or organizations imply endorsement by the US Government.

## Author contributions

G.S., T.L., K.K.L., and R.F.J. performed the experimental studies. J.W.Y. and A.A.C. supervised the work. G.S. and A.A.C. conceived the study and wrote the paper. All authors edited the paper and approved the final version.

## Competing interests

The authors declare no competing interests.
