## [Peer Review File · Nature Communications]

Reviewers' comments:

Reviewer #1 (Remarks to the Author):

The manuscript "Omicron Spike confers enhanced infectivity and interferon resistance to SARS-CoV-2 in human nasal tissue" by Shi and colleagues, aims to understand contributing factors to increased Omicron BA.1 transmissibility compared to ancestral SARS-CoV-2 variants.

While the research question is tantalising and would benefit our understanding of the global success of the Omicron variants, there are major shortcomings in the experiments presented in this manuscript and the data is not robust enough to support most of the claims. Moreover, the data are largely descriptive and link repeatedly to infer evasion/resistant to restriction factors/innate immune antiviral proteins, but no effort is made to address what these host cell factors that they claim Omicron is better able to evade actually are.

Major concerns:

- The study aims to directly compare the replicative capacity of the Omicron BA.1 variant with the ancestral SARS-CoV-2 strain USA-WA1/2020 (WA1). To achieve a direct comparison of infection kinetics between these 2 variants, it is essential that the amount of particles in the infection inoculum are equivalent. Equivalent input will allow a direct comparison of the performance of each variant in each respective infection model. The authors endeavour to achieve this by titrating virus stocks on VeroE6 cells. However, given that Omicron have shifted tropism compared to previous variants, this cell-based assay to infer virus stock titres is heavily flawed as it will skew the readouts. This will ultimately result in under- or overestimation of the viral input and skew the results and affect the overall replication dynamics of each virus. The viral genome copies in the input for the experiments should be determined by, for example, determining copies of Orf1a. It is absolutely imperative that the inoculi are comparable between viruses, and lack of appropriate normalization makes any comparative interpretation of the data between variants problematic. Indeed, the enhanced replication and cell binding reported for Omicron BA.1 may simply reflect more initial inoculi of this variant, given the way the titre was determined, as outlined above.
- WA.1 as the comparator virus does not appear to be the appropriate choice to understand increased BA.1 transmissibility. This isolate has been recovered very early in the pandemic and is a Pango lineage A strain which has not acquired the D614G mutation in SARS-CoV-2 spike yet. All dominant lineages since summer 2020 are Pango lineage B viruses and contain the D614G mutation. This mutation has been shown to be important for infectivity (including improved infectivity of primary cells) and evasion of innate immune responses (such as IFITMs) (<https://doi.org/10.7554/eLife.65365> <https://doi.org/10.1038/s41586-020-2895-3>). Choosing WA1 as a comparator for BA.1 does thus not allow to directly discern Omicron-specific replication/transmission advantages, but would more broadly unveil differences between the ancestral SARS-CoV-2 strain and variants better adapted to the human host. The use of WA1 omits important adaptations in spike that have occurred already in variants pre-Omicron. A direct comparison with a previous circulating VOC, such as Delta, would have been more appropriate. This is a significant problem with the study.

- Shi et al focus in infection of primary airway epithelial models to understand SARS-CoV-2 biology. Performing experiments in relevant primary models can indeed reveal important viral characteristics that use of cell lines will mask. However, considerations when choosing the primary cell model need to be made. It is known that ACE2 expression is substantially upregulated by ALI-differentiation of epithelial cells (<https://doi.org/10.1128/jvi.79.23.14614-14621.20> <https://doi.org/10.1016/j.heliyon.2023.e14383>). Submerged primary airway epithelial cells have previously been shown to express no or very low levels of the SARS-CoV-2 entry receptor ACE2. In Figure 1 (also relevant for Fig. 3), where viral replication is addressed over time, replication only increases 4fold over time (24-72h), which is not supporting robust replication in the submerged cells (for comparison, replication in ALI differentiated cells shows log₁₀ increases in replication <https://www.nature.com/articles/s41564-022-01143-7> <https://www.nature.com/articles/s41586-022-04474-x>). ACE2 protein levels on the cell surface of submerged primary cell cultures should be shown and infection experiments should be performed in the presence or absence of remdesivir over time to address whether the observed increase in viral Orf1a transcripts is reflecting viral replication.

- The authors propose that rather than relying on cathepsins, MMPs facilitate Omicron entry into primary nasal epithelial cells. It is intriguing to contemplate that Omicron entry might be mediated via a novel entry route. ADAMs and MMPs have previously been reported to support SARS-CoV-2 VOC (including Omicron) entry into airway epithelial cells (<https://www.embopress.org/doi/full/10.15252/embr.202154305>) similarly, incyclinide has previously been reported to inhibit SARS-CoV-2 entry (<https://www.ncbi.nlm.nih.gov/pmc/articles/PMC9858505/>). However, more robust experimental evidence is needed to determine the contribution of MMPs. The proposed mechanism is entirely dependent on the specificity (How specific is Incyclinide for MMP inhibition at the chosen concentration?) and potency of the chosen inhibitors in Fig. 2G. For example, are the authors confident that E64d retained its activity in the ALI-nasal cultures? Bafilomycin A has been shown to efficiently inhibit replication of ancestral SARS-CoV-2, treatment of WA1 infected cells with the inhibitor would have corroborated whether the enhancement seen during BA.1 infection can be attributed to the inhibitor working. Regarding the involvement of MMP14/16, what are the authors basing their hypotheses on? Do they have data to show the involvement of these specific MMPs?

- The manuscript heavily relies on single time point measurements, which sometimes seem arbitrarily chosen (24h or 48h). Several timepoint measurements are needed to assess the replication kinetics of each variant (i.e. is WA1 struggling in primary epithelial cells and replicating poorly, as it might be expected of a very early ancestral variant? Poor replication would make it hard to directly compare the results to BA.1.). I appreciate that primary epithelial cultures are a resource intense infection model, however, even if not enough material is available to prepare cell lysates at each time point, virus release from the apical side of the ALI-cultures could have been measured over time from an infected sample. Similarly, this would allow the authors to truly discern the “replicative potential” of the variants.

- The data presented is shown as normalized data to a particular timepoint (for example Fig. 1C) or variant (for example. Fig. 2G). This normalization does not allow the reader to discern the absolute levels of replication. Since this manuscript aims to understand differences between variants, normalizing infection levels within each variant to 1, does not allow direct comparisons between variants or allow to understand the magnitude of changes in replication or IFN sensitivity that are reported. In Fig1 D and E, 2 timepoints are shown (48h and entry/adherence at 1h) from the data, it is impossible to tell whether between those 2 timepoints any replication has occurred. Similarly, the fold change reduction of viral

replication through interferon will depend on how much replication is differing from the baseline (incoming virus). These values are likely very different between variants, and as such fold changes mask the actual effects occurring. Data should be shown either in a non-normalized manner, or even better viral replication should be reported as absolute values based on a qPCR assay that allows quantification of viral replication through quantification of genome copies. This would for example allow to discern whether viral replication between 1h (adherence of the virus) and 48h (after several rounds of viral replication) has occurred.

- Can the authors comment on whether their mCherry-RG mutant viruses (with mCherry linked to N protein), would already be detectable at input, by the notion that N is highly abundant in virus particles, and thus should be mCherry (Fig. 2E)?
- The data shown in Fig. 1B and C seems to have been normalized differently. B seems to be showing both variants normalized to WA1 while C seems to be normalized to each respective variant. Can the authors clarify?
- Why was only one donor used for the SAEC experiments? Infections should be performed with independent donors. Does “independent RT-qPCR runs” mean one experiment has been assayed by RT-PCR repeatedly, or that independent experiments have been assayed? When FFUs are reported, do “three independent infections of Vero cells” refer to virus harvested from 3 independent nasal cell infections were used or the supernatants from 1 experiment tested 3 times? It should be stated clearly whether independent infections have been performed.
- The way the figure legends are reported, it is difficult to understand how often a particular experiment has been independently performed.

Minor concerns:

- The manuscript refers to BA.5 at some point – I assume this means BA.2
- The authors should discuss the role of antibody evasion in the dominance of BA.1 over Delta.
- IF images in Figure 1 and 2 require scale bars.
- Show ISG induction in Fig. 1 normalised to uninfected cells.
- Ifnb in Fig 1 should be IFNB. Human gene names are reported in upper case letters and italicised.
- In the introduction, “...BA.1 contained a plethora of unique mutations, and its descendants (including the BA.2, BA.4, BA.5, BQ.1, and XBB subvariants)...” needs to be corrected BA.2 etc are not derived from BA.1, they share a common ancestor.
- Can the authors clarify how long were ALI epithelial cultures incubated with buffer on the apical side to recover released virus?

Reviewer #2 (Remarks to the Author):

The omicron variant of SARS-CoV-2 emerged in late 2021 and quickly supplanted the previously dominant Delta variant worldwide. The omicron variant possesses an extraordinarily high resistance to antibody-mediated neutralization, primarily due to numerous mutations in the spike protein (S). This resistance allows the omicron variant to undermine preexisting immunity. Additionally, a shift in the preferred host cell entry route has enabled the omicron variant to replicate more efficiently in the upper respiratory tract, thereby increasing its transmissibility. In this study, Shi and colleagues investigated omicron variant infection of nasal epithelial cells to uncover the molecular mechanisms responsible for its heightened replication potential in the upper respiratory tract. The findings indicate that the enhanced infectivity of nasal epithelial cells can be attributed to the cell entry step and specifically mapped to the S protein of the omicron variant. Furthermore, the study reveals that the entry of omicron S into nasal epithelial cells is facilitated by matrix metalloproteinases, and this entry strategy allows the variant to evade interferon-inducible factors that target viral entry.

Overall, this study provides valuable insights into the molecular mechanisms underlying the increased replication capacity of the omicron variant in the upper respiratory tract. The manuscript is original, well-written and presents the data clearly. However, it is unfortunate that no attempt was made to identify the cellular factor(s), possibly ISG(s), responsible for the IFN-mediated inhibition of WA1 but not omicron infection in nasal epithelial cells. Such data are necessary to gain a comprehensive understanding of the molecular mechanisms behind the omicron variant's enhanced replication in upper respiratory tract cells and would justify publication in a high-impact journal.

Specific points:

1. Introduction, line 7: The authors state that omicron S displays similar ACE2 affinity compared to D614G S and cite the studies by Han et al. and Wang et al. However, there are several studies that have found increased ACE2 affinity for omicron S (e.g., PMIDs: 35228716 and 35714668). Importantly, Wang et al. also report slightly higher ACE2 affinity for omicron S (BA.1, BA.1.1, BA.2, BA.2.12.1, BA.4/5) compared to D614G S. Therefore, this paragraph should be revised and also cite studies that report increased ACE2 affinity for omicron S.

2. It would be straightforward to determine which IFN-induced factors restrict the entry of WA1 but not omicron in nasal epithelial cells. For instance, comparing the ISG expression profiles of IFN-treated and control-treated cells could identify candidate ISGs. Subsequently, selected ISGs could be inhibited (e.g., using siRNA/shRNA), and the impact on WA1 and omicron infection could be analyzed.

3. Key findings should be confirmed with currently circulating omicron subvariants (i.e., XBB.1.5 or XBB.1.16).

Reviewer #3 (Remarks to the Author):

Omicron Spike confers enhanced infectivity and interferon resistance to SARS-CoV-2 in human nasal tissue

Shi, Li et al.,

The authors present data showing the Omicron variant of SARS-CoV-2 increases infectivity in primary upper respiratory tract tissue. The entry of Omicron, unlike other variants, happens independently of TMPRSS2 and uses metalloproteinases to activate membrane fusion via the Spike. The authors show that Omicron is resistant to IFN-induced restriction following cell binding.

Comments:

1) The authors use Vero E6-TMPRSS2 cells to propagate the SARS-CoV-2 and then use Vero E6 cells to calculate infectious titers. The titers derived from Vero E6 infections are used to calculate the MOI for the infection studies using WA1, BA.1, BA.2. This approach may confound the MOI calculation as the Omicron variant is less independent on TMPRSS2 and cathepsins for infectivity (Figure 2G). The authors should titer the viruses using Vero E6-TMPRSS2 to confirm the titres are not affected by the absence/presence of TMPRSS2 in Omicron.

2) Figure 1E: Virus attachment to cells show a 30-fold higher value for BA.1 compared to WA1. This is puzzling. It indicates that BA.1 has 30-fold higher infectious particles in the inoculant. If this is true, the higher FFUs seen in BA.1 at 24hpi are not surprising.

3) Related to 2), the authors should rule out that the data shown in Figure 1D, 2B-2D, 2F, 3C are not just a reflection of higher infectious input in the inoculant of BA.1 or BA.2.

4) P6 line3-5

“It is possible that an improved capacity for Omicron Spike to bind heparan sulfate and neuropilin 1 may negatively interfere with processing by TMPRSS2 and steer processing towards MMPs such as MMP14 and MMP16. “ It may be informative to test the neuropilin 1 antagonist (EG00229) on its effect on Omicron.

5) The authors postulate that IFN induced proteins fail to block Omicron due to Omicron’s ability to enter cells independent of TMPRSS2. Do they have an indication of what IFN induced proteins are being bypassed by Omicron?

Minor points:

1) Does Omicron overcome restriction in cells over-expressing IFITMs?

Typos:

P2 line3 – characterization of Omicron....

P5 last line and elsewhere in text (P6 line4) – neuropilin 1 not neuropilin 1.

Reviewers' comments and authors' rebuttal:

Reviewer #1 (Remarks to the Author):

The manuscript “Omicron Spike confers enhanced infectivity and interferon resistance to SARS-CoV-2 in human nasal tissue” by Shi and colleagues, aims to understand contributing factors to increased Omicron BA.1 transmissibility compared to ancestral SARS-CoV-2 variants.

While the research question is tantalising and would benefit our understanding of the global success of the Omicron variants, there are major shortcomings in the experiments presented in this manuscript and the data is not robust enough to support most of the claims. Moreover, the data are largely descriptive and link repeatedly to infer evasion/resistant to restriction factors/innate immune antiviral proteins, but no effort is made to address what these host cell factors that they claim Omicron is better able to evade actually are.

We thank the reviewer for their positive appraisal of our manuscript, namely the fact that the research question is of extraordinary interest and the findings of which would help explain the dominance and spread of Omicron globally. As we outline below, we have revised our manuscript and added a significant amount of new data to support our primary claim—that Omicron BA.1 and other Omicron sublineages exhibit an increased capacity to infect primary human nasal epithelia, relative to ancestral D614G and Delta, and this is the product of Spike mutations specific to the Omicron lineage. We also provide new data showing that the block of WA1 and Delta in primary nasal epithelia is mediated, at least partially, by IFITM proteins. These results have significant implications for the mechanistic understanding of how BA.1 came to dominate the previous variant of concern, Delta, and it provides important insight into the mechanisms shaping the evolution of future coronaviruses with pandemic potential.

Major concerns:

- The study aims to directly compare the replicative capacity of the Omicron BA.1 variant with the ancestral SARS-CoV-2 strain USA-WA1/2020 (WA1). To achieve a direct comparison of infection kinetics between these 2 variants, it is essential that the amount of particles in the infection inoculum are equivalent. Equivalent input will allow a direct comparison of the performance of each variant in each respective infection model. The authors endeavour to achieve this by titrating virus stocks on VeroE6 cells. However, given that Omicron have shifted tropism compared to previous variants, this cell-based assay to infer virus stock titres is heavily flawed as it will skew the readouts. This will ultimately result in under- or overestimation of the viral input and skew the results and affect the overall replication dynamics of each virus. The viral genome copies in the input for the experiments should be determined by, for example, determining copies of Orf1a. It is absolutely imperative that the inoculi are comparable between viruses, and lack of appropriate normalization makes any comparative interpretation of the data between variants problematic. Indeed, the enhanced replication and cell binding reported for Omicron BA.1 may simply reflect more initial inoculi of this variant, given the way the titre was determined, as outlined above.

We agree that it is important to rule out that differing inoculums were partially or fully responsible for the difference in infection outcome, and another reviewer also pointed this out. Originally, we titered all of our viruses on Vero E6-TMPRSS2 cells to obtain infectious titers

(the only way to infer infectious viral particle quantity). This is the gold standard used in most publications reporting on the virological descriptions of SARS-CoV-2 variants. For example, see this recent Nature paper published in August describing the characteristics of the latest Omicron sublineages (Adettia et al, “Neutralization, effector function, and immune imprinting of Omicron variants.” PMID: 35228716). In this article, they state in the methods that “All viruses were propagated and titered on Vero E6-TMPRSS2 cells,” which is exactly how we propagated and titered all of the viruses used in our study.

Nonetheless, we went ahead and performed additional experiments in order to exclude that our titration practice was introducing an artifact whereby inoculants contained different amounts of virus particles, therefore biasing our results. We have added new results to Supplemental Figure 1 showing that the inoculants used for WA1 and BA.1 result in comparable infection of HEK293T-ACE2 cells (Supplemental Figure 1E) and VeroE6-TMPRSS2 cells (Supplemental Figure 1F), indicating that differences in virus input quantity cannot explain the enhanced ability of BA.1 to adhere to and infect nasal cells. Also, we showed in our initial submission that BA.1 exhibits poor replication in primary lung cells compared to WA1, and this result would not be expected if the inoculant of BA.1 was simply higher than that of WA.1.

Furthermore, based on this reviewer’s suggestion, we also chose to titer our virus using an alternative method (viral qPCR) instead of measuring infectious titers by focus forming units assay in Vero E6-TMPRSS2. When we reperformed a side by side infection of WA1 and BA.1 using new titers calculated by viral qPCR, we observed that BA.1 still infected nasal cells significantly better than WA1 (Supplemental Figure 1G). Additionally, we reperformed infections with recombinant WA1-mCherry virus encoding WA1 Spike, Delta Spike, or BA.1 Spike using new titers calculated by viral qPCR. The results were identical to those obtained through the traditional titering method (infectious virus particle enumeration on Vero E6-TMPRSS2 cells) (Supplemental Figure 2D and 2E).

Overall, our data indicate that BA.1 Spike confers a selective infectivity advantage that is apparent in nasal cells but not apparent in other cell lines. Therefore, we are able to exclude that differences in the inoculants are responsible for the perceived increased in nasal cell infectivity of Omicron. Instead, our results demonstrate that Omicron has evolved a superior capacity to infect primary nasal epithelia, meaning that it has adapted to nasal cell-intrinsic host factors to expedite its entry process in this tissue.

- WA.1 as the comparator virus does not appear to be the appropriate choice to understand increased BA.1 transmissibility. This isolate has been recovered very early in the pandemic and is a Pango lineage A strain which has not acquired the D614G mutation in SARS-CoV-2 spike yet. All dominant lineages since summer 2020 are Pango lineage B viruses and contain the D614G mutation. This mutation has been shown to be important for infectivity (including improved infectivity of primary cells) and evasion of innate immune responses (such as IFITMs) (<https://doi.org/10.7554/eLife.65365> <https://doi.org/10.1038/s41586-020-2895-3>). Choosing WA1 as a comparator for BA.1 does thus not allow to directly discern Omicron-specific replication/transmission advantages, but would more broadly unveil differences between the ancestral SARS-CoV-2 strain and variants better adapted to the human host. The use of WA1 omits important adaptations in spike that have occurred already in variants pre-Omicron. A direct comparison with a previous circulating VOC, such as Delta, would have been more appropriate. This is a significant problem with the study.

We agree with the reviewer's suggestion to compare BA.1 with a more proximal preexisting variant, such as D614G or the Delta variant of concern. Therefore, in addition to WA1, each figure now shows BA.1 compared to D614G, Delta, or both. The new results provided in this revision demonstrate that D614G and Delta infect primary nasal epithelia to a lesser extent than BA.1, supporting our original conclusion that increased nasal cell infectivity evolved in the Omicron ancestor and was not already a feature of the D164G variant or the Delta variant of concern. Furthermore, we provide evidence that WA1 encoding WA1 Spike or Delta Spike are sensitive to Amphotericin B, which counteracts the antiviral activity of endogenous IFITM proteins. In contrast, BA.1 Spike-mediated infection is relatively insensitive to Amphotericin B. Thus, the mutations present in Omicron Spike confer at least partial resistance to IFITM proteins in primary nasal epithelia, while mutations present in Delta Spike (which contains the D614G mutation) do not. These results are now found in Figure 5 and Supplemental Figure 5 of the revised manuscript. We believe these results are significant because whether IFITM perform antiviral or proviral functions during SARS-CoV-2 infection has remained controversial. Our findings demonstrate that IFITM are playing an antiviral role in this very relevant primary nasal tissue and limit infection of early/ancestral SARS-CoV-2. We thank the reviewer for these suggestions because we believe the article is now significantly improved, more insightful, and more appropriate for Nature Communications.

- Shi et al focus in infection of primary airway epithelial models to understand SARS-CoV-2 biology. Performing experiments in relevant primary models can indeed reveal important viral characteristics that use of cell lines will mask. However, considerations when choosing the primary cell model need to be made. It is known that ACE2 expression is substantially upregulated by ALI-differentiation of epithelial cells (<https://doi.org/10.1128/jvi.79.23.14614-14621.20> <https://doi.org/10.1016/j.heliyon.2023.e14383>). Submerged primary airway epithelial cells have previously been shown to express no or very low levels of the SARS-CoV-2 entry receptor ACE2. In Figure 1 (also relevant for Fig. 3), where viral replication is addressed over time, replication only increases 4fold over time (24-72h), which is not supporting robust replication in the submerged cells (for comparison, replication in ALI differentiated cells shows log₁₀ increases in replication <https://www.nature.com/articles/s41564-022-01143-7> <https://www.nature.com/articles/s41586-022-04474-x>). ACE2 protein levels on the cell surface of submerged primary cell cultures should be shown and infection experiments should be performed in the presence or absence of remdesivir over time to address whether the observed increase in viral Orf1a transcripts is reflecting viral replication.

We agree that our use of submerged primary nasal epithelial cells in some of our figures raised questions over whether these cells are capable of supporting productive infection and virus spread over time. To directly address these questions, we measured ACE2 levels in submerged primary nasal epithelial cells with anti-ACE2 and immunofluorescence confocal microscopy. Detection of ACE2 is now provided in Figure 1A, next to the immunofluorescence results with DAPI and anti-acetylated tubulin. As the reviewer states, ACE2 levels are lower in the submerged epithelia compared to cells differentiated at the liquid-air interface (anti-ACE2 immunofluorescence is also now provided in nasal air-liquid interface tissue in Figure 2A). By providing the raw, unnormalized results of viral ORF1a qPCR from infected cells, we show that air-liquid interface tissue supported more than 100-fold higher infection than submerged nasal epithelial cells (Supplemental Figure 1B and Supplemental Figure 2A). To address the

replicative potential of SARS-CoV-2 strains in submerged nasal epithelia, we inoculated these cells with WA1 and checked the infectious potential of supernatants collected from cell culture medium at 24, 48, and 72 hours post-inoculation. We found that cell culture supernatants contained infectious virus, as evidenced by productive infection of permissive Vero E6-ACE2-TMPRSS2 cells by viral ORF1a qPCR and anti-N immunofluorescence staining (Supplemental Figure 1A).

We also used remdesivir to demonstrate that our measurements of ORF1a RNA levels in inoculated cells reflect genuine SARS-CoV-2 replication following inoculation (Figure 1E). Therefore, while submerged nasal epithelia are less permissive to SARS-CoV-2 than tissue at the air-liquid interface, they nonetheless support the full infectious life cycle of the virus and are appropriate for comparative studies between Omicron lineages, previous variants of concern, and early/ancestral SARS-CoV-2. We believe our description of the two primary models used in our study will be instructive to many other labs working on SARS-CoV-2 virology.

- The authors propose that rather than relying on cathepsins, MMPs facilitate Omicron entry into primary nasal epithelial cells. It is intriguing to contemplate that Omicron entry might be mediated via a novel entry route. ADAMs and MMPs have previously been reported to support SARS-CoV-2 VOC (including Omicron) entry into airway epithelial cells (<https://www.embopress.org/doi/full/10.15252/embr.202154305>) similarly, incyclinide has previously been reported to inhibit SARS-CoV-2 entry (<https://www.ncbi.nlm.nih.gov/pmc/articles/PMC9858505/>). However, more robust experimental evidence is needed to determine the contribution of MMPs. The proposed mechanism is entirely dependent on the specificity (How specific is Incyclinide for MMP inhibition at the chosen concentration?) and potency of the chosen inhibitors in Fig. 2G. For example, are the authors confident that E64d retained its activity in the ALI-nasal cultures? Bafilomycin A has been shown to efficiently inhibit replication of ancestral SARS-CoV-2, treatment of WA1 infected cells with the inhibitor would have corroborated whether the enhancement seen during BA.1 infection can be attributed to the inhibitor working. Regarding the involvement of MMP14/16, what are the authors basing their hypotheses on? Do they have data to show the involvement of these specific MMPs?

We agree that verifying the activity of any inhibitor used in our studies is of paramount importance. As the reviewer points out, incyclinide was previously reported to inhibit SARS-CoV-2 entry (Chan et al., Science Advances, 2023), and we used this paper as a guide to choose the concentration of incyclinide that we used in our experiments. We saw that 10 micromolar of incyclinide resulted in strong inhibition of WA1, Delta, and BA.1, with BA.1 exhibiting more sensitivity than the other viruses, which is consistent with the findings from Chan et al. It was shown to inhibit early isolates of SARS-CoV-2 infection in Chan et al. and we used it at the same concentration that they did (10 micromolar, which was shown not to inhibit cell viability). Based on this finding, and since we are using human primary tissue in our experiments, we did not perform infections with a spectrum of different concentrations of incyclinide (as it was not feasible). To expand the functional data presented on MMPs/ADAMs usage by different variants, we tested two other inhibitors, apratastat and batimastat. Apratastat is considered an inhibitor of ADAM10, ADAM17 and MMP-13, while batimastat is capable of inhibiting MMP-1, MMP-2, MMP-9, MMP-7, MMP-3, and ADAM17. Importantly, we found that, while incyclinide inhibited both Delta and BA.1 infections in nasal epithelia, apratastat and batimastat

had a more selective effect—Delta was completely insensitive to both, while BA.1 was inhibited by 60% by both (Figure 3C). The fact that apratastat and batimastat inhibit BA.1 but not Delta indicates that they inhibit a narrower range of MMPs/ADAMs than incyclinide, and we therefore uncovered a functional distinction between the Delta and BA.1 entry pathways based on this result. Since apratastat was developed for clinical use as an ADAM17 inhibitor, and batimastat can also target ADAM17, our results suggest that BA.1 Spike may utilize ADAM17 for its processing in primary nasal epithelia, while Delta Spike does not. However, apratastat and batimastat may also inhibit ADAM10 and other proteases unknown to us, in addition to ADAM17. We now discuss how our experiments reveal a possible involvement of ADAM17, ADAM10 and other metalloproteinases in BA.1 Spike-mediated entry in nasal tissue. We have removed the specific mentioning of MMP-14/16 since we do not provide any direct evidence that those MMPs are implicated in BA.1 entry.

Concerning E64d, we now explicitly state in the text that E64d treatment reduced WA.1 infection by 25%, which indicates E64d exhibits activity in this tissue type at the concentration used. Further attesting to E64d activity is the fact that it significantly boosts both Delta and BA.1 infection. These results reveal important distinction between early/ancestral SARS-CoV-2 and later variants of concern. Together with the results from camostat, these results indicate that Delta is using a cathepsin-independent, TMPRSS2-dependent entry pathway. In contrast, the entry pathway for BA.1 is cathepsin-independent and TMPRSS2-independent. Importantly, WA1 is able to use cathepsin-dependent and TMPRSS2-dependent entry pathways (Figure 3A). Concerning bafilomycin A1, we now provide new results whereby WA1, Delta, and BA.1 were pretreated with bafilomycin A1 and the impact on infection was measured. In our initial submission, we showed that BA.1 was boosted, but not inhibited, by bafilomycin A1. We now show that, in contrast to BA.1, both WA1 and Delta infections are reduced by bafilomycin A1 (Figure 3A), and this finding was recapitulated using recombinant WA1 viruses encoding Delta Spike or BA.1 Spike (Figure 3B). These results significantly extend the meaningfulness of the figure, since they imply that WA1 and Delta can productively enter nasal epithelia via an endocytic entry route while BA.1 cannot. Given that some published studies and preprints claim that BA.1 enters human cells through an endocytic route, our findings showing that BA.1 does not enter via an endocytic route in primary nasal epithelia are of high significance to the scientific and lay communities, and we believe this is one of the reasons why our study should be considered at Nature Communications.

- The manuscript heavily relies on single time point measurements, which sometimes seem arbitrarily chosen (24h or 48h). Several timepoint measurements are needed to assess the replication kinetics of each variant (i.e. is WA1 struggling in primary epithelial cells and replicating poorly, as it might be expected of a very early ancestral variant? Poor replication would make it hard to directly compare the results to BA.1.). I appreciate that primary epithelial cultures are a resource intense infection model, however, even if not enough material is available to prepare cell lysates at each time point, virus release from the apical side of the ALI-cultures could have been measured over time from an infected sample. Similarly, this would allow the authors to truly discern the “replicative potential” of the variants.

We agree that our infection results using a single endpoint made it hard to appreciate whether different viruses exhibited differential replicative potential. We decided to expand the results obtained using our novel, recombinant WA1-mCherry viruses encoding WA1 Spike, Delta

Spike, or BA.1 Spike for this purpose. As was apparent in our initial submission, some infection mediated by WA1 Spike and Delta Spike was detected at 24 hours post infection, while an ample degree of infection by BA.1 Spike was detected at the same time point. We now provide assessment of earlier time points post infection (6, 9, and 12 hours post infection). Infection by WA1 (BA.1 Spike) was detected as early as 6 hours post infection, with virus growth apparent at 9 and 12 hours post infection. However, infection by WA1 (Delta Spike) only became detectable at 12 hours post infection, with only a few cells displaying mCherry (Supplemental Figure 2F). These results can be considered as the most direct means possible to assess replicative potential of different variants, and they highlight the rapid entry and spread mediated by BA.1 Spike compared to Delta Spike.

- The data presented is shown as normalized data to a particular timepoint (for example Fig. 1C) or variant (for example. Fig. 2G). This normalization does not allow the reader to discern the absolute levels of replication. Since this manuscript aims to understand differences between variants, normalizing infection levels within each variant to 1, does not allow direct comparisons between variants or allow to understand the magnitude of changes in replication or IFN sensitivity that are reported. In Fig1 D and E, 2 timepoints are shown (48h and entry/adherence at 1h) from the data, it is impossible to tell whether between those 2 timepoints any replication has occurred. Similarly, the fold change reduction of viral replication through interferon will depend on how much replication is differing from the baseline (incoming virus). These values are likely very different between variants, and as such fold changes mask the actual effects occurring. Data should be shown either in a non-normalized manner, or even better viral replication should be reported as absolute values based on a qPCR assay that allows quantification of viral replication through quantification of genome copies. This would for example allow to discern whether viral replication between 1h (adherence of the virus) and 48h (after several rounds of viral replication) has occurred.

As described above, we have now included results with remdesivir to confirm that WA1, Delta, and BA.1 replicate in submerged nasal epithelia (Figure 1E) and fluorescent analysis of recombinant WA1 (WA1 Spike), WA1 (Delta Spike), and WA1 (BA.1 Spike) in nasal air-liquid interface at multiple time points (Figure 2G and Supplemental Figure 2F). Together, these confirm that our viruses exhibit replicative potential in both of our primary nasal tissue systems. In order to render our normalized data more transparent, we now include infection results where WA1 and BA.1 were added to nasal monolayers or ALI and the absolute copy number of ORF1a was calculated. These results are now found in Supplemental Figure 1B and Supplemental Figure 2A, and the data demonstrate that BA.1 infection results in ORF1a copy numbers that are approximately two orders of magnitude greater than WA1.

It would be too resource intensive to reperform all of our infections so that we could replot all of our data in terms of ORF1a copy number. We believe that providing these examples documenting non-normalized ORF1a make all of the rest of the figures interpretable as normalized graphs, since BA.1 is found in every figure where two or more variants were compared with one another.

Furthermore, it is important to consider that some of our data is not presented in a normalized manner. The focus forming units graphs, which measure the production of infectious virus particles from inoculated nasal epithelia, are presented as FFU per mL and are not normalized (this was intentional for the sake of transparency). We believe it is clear when we are presenting

normalized or nonnormalized data because the Y-axis of the figure states “Relative” to indicate that data is normalized. Exactly how data is normalized is always described in the figure caption.

- Can the authors comment on whether their mCherry-RG mutant viruses (with mCherry linked to N protein), would already be detectable at input, by the notion that N is highly abundant in virus particles, and thus should be mCherry (Fig. 2E)?

The recombinant viruses were constructed by linking mCherry to N protein via a 2A peptide, which causes ribosome skipping and is a widely used method to ensure independent translation of the N and mCherry proteins. Therefore, mCherry is not fused to N protein, but rather, the two proteins are translated independently from one another. With the newly added data in Supplemental Figure 2F, showing virus spread of WA1-mCherry based recombinant viruses, it is apparent that mCherry signal increases over time, especially for the WA1(BA.1 Spike) virus. mCherry signal after inoculation with WA1 (BA.1) is hardly detectable at 6 hours post inoculation and increases steadily between 6-24 hours. Between 0 and 4 hours post inoculation, none of the recombinant viruses produce detectable mCherry signal. Therefore, the mCherry is not detectable at input and is only visualized following virus replication.

- The data shown in Fig. 1B and C seems to have been normalized differently. B seems to be showing both variants normalized to WA1 while C seems to be normalized to each respective variant. Can the authors clarify?

We are unsure what the reviewer is referring to, since Figure 1 did not seem to contain data sets that were normalized differently. Perhaps the author meant to speak about the data that is now found in Figure 2. In this figure, the ORF1a qPCR data in Figure 2B is normalized such that WA1 is set to 1 and all other measurements are made relative to 1. However, the focus-forming units data in Figure 2C is presented in a nonnormalized matter to make the result as transparent as possible. That is, cells inoculated with BA.1 and BA.2 produce more infectious virus particles compared to cells inoculated with WA1. This is described in the Figure 2 caption.

- Why was only one donor used for the SAEC experiments? Infections should be performed with independent donors. Does “independent RT-qPCR runs” mean one experiment has been assayed by RT-PCR repeatedly, or that independent experiments have been assayed? When FFUs are reported, do “three independent infections of Vero cells” refer to virus harvested from 3 independent nasal cell infections were used or the supernatants from 1 experiment tested 3 times? It should be stated clearly whether independent infections have been performed.

We have now updated Figure 1C so that three donors (pooled) were used for the hSAEC experiments. We believe pooling the different donors is the best way to accommodate donor-to-donor variability, and our nasal tissue work involved pooled donors as well. The inclusion of three pooled donors did not change our conclusion—BA.1 does not exhibit an increased capacity relative to WA1 to infect these primary lung cells, suggesting that the increased capacity for BA.1 to infect nasal cells involves nasal cell-intrinsic host factors. We have updated all figure legends/captions to ensure that how we define experimental replicates is clear. All figure plots include dots/symbols to indicate the experimental result for each replicate, and in general, all

replicates are defined as independent virus inoculations in independent tissue culture wells (biological replicates).

- The way the figure legends are reported, it is difficult to understand how often a particular experiment has been independently performed.

We have updated all figure legends/captions to ensure that how we define experimental replicates is clear. All figure plots include dots/symbols to indicate the experimental result for each replicate, and in general, all replicates are defined as independent virus inoculations in independent tissue culture wells (biological replicates).

Minor concerns:

- The manuscript refers to BA.5 at some point – I assume this means BA.2

We thank the reviewer for identifying this typo, and it has been corrected.

- The authors should discuss the role of antibody evasion in the dominance of BA.1 over Delta.

We thank the reviewer for this suggestion. We added additional text to the Discussion to indicate that antibody evasion has contributed to the evolution of Omicron and contributed to its dominance in the post-vaccination era. At line 422, we state “While some mutations in Omicron Spike may have conferred Omicron with superior evasion of vaccine-elicited neutralizing antibodies compared to the Delta variant, our findings indicate that mutations in Omicron Spike also allow for efficient seeding of infection in the nasal epithelium and resistance to the interferon-induced antiviral state therein. Our findings provide mechanistic insight into the basis for the efficient transmission and persistence of Omicron in vaccinated and unvaccinated human populations. Future efforts aimed at understanding how individual Spike mutations contribute to these different phenotypes may enable a better understanding of the selective forces driving their emergence.”

- IF images in Figure 1 and 2 require scale bars.

Scale bars are now included, and they are defined in the figure captions. Thank you.

- Show ISG induction in Fig. 1 normalised to uninfected cells.

We thank the reviewer for this great suggestion. We have included qPCR results showing the induction of *IFITM3* in nasal ALI inoculated with different virus variants (WA1, D614G, Delta, BA.1, and BQ.1). Compared to a non-inoculated control (in which *IFITM3* mRNA levels were set to 1), all virus variants induced *IFITM3*, with Omicron BA.1 and BQ.1 inducing the most *IFITM3*. These data can now be found in Supplemental Figure 2B.

- *Ifnb* in Fig 1 should be IFNB. Human gene names are reported in upper case letters and italicised.

Corrected. Thank you.

- In the introduction, "...BA.1 contained a plethora of unique mutations, and its descendants (including the BA.2, BA.4, BA.5, BQ.1, and XBB subvariants)..." needs to be corrected BA.2 etc are not derived from BA.1, they share a common ancestor.

We thank the reviewer for pointing this out, and it has been corrected.

- Can the authors clarify how long were ALI epithelial cultures incubated with buffer on the apical side to recover released virus?

200 uL of D-PBS was added to the apical side of ALI for 20 minutes at 37 degrees, and the D-PBS was then harvested to recover released virus. We added additional text to the methods section at line 749 to make this clear.

Reviewer #2 (Remarks to the Author):

The omicron variant of SARS-CoV-2 emerged in late 2021 and quickly supplanted the previously dominant Delta variant worldwide. The omicron variant possesses an extraordinarily high resistance to antibody-mediated neutralization, primarily due to numerous mutations in the spike protein (S). This resistance allows the omicron variant to undermine preexisting immunity. Additionally, a shift in the preferred host cell entry route has enabled the omicron variant to replicate more efficiently in the upper respiratory tract, thereby increasing its transmissibility. In this study, Shi and colleagues investigated omicron variant infection of nasal epithelial cells to uncover the molecular mechanisms responsible for its heightened replication potential in the upper respiratory tract. The findings indicate that the enhanced infectivity of nasal epithelial cells can be attributed to the cell entry step and specifically mapped to the S protein of the omicron variant. Furthermore, the study reveals that the entry of omicron S into nasal epithelial cells is facilitated by matrix metalloproteinases, and this entry strategy allows the variant to evade interferon-inducible factors that target viral entry.

Overall, this study provides valuable insights into the molecular mechanisms underlying the increased replication capacity of the omicron variant in the upper respiratory tract. The manuscript is original, well-written and presents the data clearly. However, it is unfortunate that no attempt was made to identify the cellular factor(s), possibly ISG(s), responsible for the IFN-mediated inhibition of WA1 but not omicron infection in nasal epithelial cells. Such data are necessary to gain a comprehensive understanding of the molecular mechanisms behind the omicron variant's enhanced replication in upper respiratory tract cells and would justify publication in a high-impact journal.

We are glad that the reviewer found our manuscript original, valuable, and well-written. We have remedied the fact that we did not include data on which possible ISGs may be contributing to the differences in infection observed between WA1, Delta, and BA.1. Please see our description below about how we used Amphotericin B to show that IFITM-mediated restriction is, at least partially, responsible for the differences in infectivity between WA1, Delta, and BA.1.

Specific points:

1. Introduction, line 7: The authors state that omicron S displays similar ACE2 affinity compared to D614G S and cite the studies by Han et al. and Wang et al. However, there are several studies that have found increased ACE2 affinity for omicron S (e.g., PMIDs: 35228716 and 35714668). Importantly, Wang et al. also report slightly higher ACE2 affinity for omicron S (BA.1, BA.1.1, BA.2, BA.2.12.1, BA.4/5) compared to D614G S. Therefore, this paragraph should be revised and also cite studies that report increased ACE2 affinity for omicron S.

We thank the reviewer for these suggested improvements. We have revised the Introduction paragraph in question and added the recommended citations at line 52: “Furthermore, some studies have shown that Omicron Spike displays increased affinity for human ACE2 compared to the early D614G variant^{23,27,28}, but this was not supported by others²⁹.”

2. It would be straightforward to determine which IFN-induced factors restrict the entry of WA1 but not omicron in nasal epithelial cells. For instance, comparing the ISG expression profiles of IFN-treated and control-treated cells could identify candidate ISGs. Subsequently, selected ISGs could be inhibited (e.g., using siRNA/shRNA), and the impact on WA1 and omicron infection could be analyzed.

We appreciate the viewpoint of the reviewer and agree it would be important to know what IFN-induced factors restrict the entry of WA1 and Delta, but not Omicron. We believe that fully addressing the IFN-induced genes responsible for restriction of WA1 and Delta (and which fail to block Omicron) would require a genetic screening approach. Given that we use entirely primary human tissue for our work, this is outside the scope of this study. However, we decided to take advantage of the fact that Amphotericin B was previously shown to counteract the antiviral properties of IFITM2 and IFITM3 in cells. We have now added new data showing that Amphotericin B pre-treatment boosted infection of nasal air-liquid interface tissue by WA1 (WA1 Spike) and WA1 (Delta Spike) by approximately 7-fold, suggesting that IFITM proteins exhibit antiviral properties in this human primary tissue. Interestingly, infection by WA1 (BA.1 Spike) was boosted to a lesser extent by Amphotericin B (less than 2-fold). We employed recombinant viruses encoding mCherry to assess virus infection at multiple time points to come to these conclusions. Collectively, these results suggest that BA.1 Spike confers virus with partial resistance to antiviral IFITM proteins in nasal tissue. Therefore, it appears that IFITM proteins play a role in the limited virus replication exhibited by WA1 and Delta in human nasal tissue, while BA.1 Spike evolved at least partial resistance to IFITM proteins. Moreover, the significance of these results can be appreciated further when one considers that the role of IFITM proteins during SARS-CoV-2 infection has been controversial—some studies have raised the possibility that endogenous IFITM proteins are dependency factors for SARS-CoV-2 in certain cell lines (i.e. they are not antiviral). Our data suggest that endogenous IFITM proteins most likely perform inhibitory roles against early/ancestral SARS-CoV-2 in primary human nasal tissue and indicate that resistance to IFITM proteins may have contributed to the dominance of Omicron lineages in human populations. These results can now be found in Figure 5 and Supplemental Figure 5.

3. Key findings should be confirmed with currently circulating omicron subvariants (i.e., XBB.1.5 or XBB.1.16).

We have now included new results of BQ.1 (a BA.5 subvariant) and XBB (a BA.2 subvariant) infections in nasal ALI. BQ.1 was widespread in early 2023 while XBB is the ancestor of the currently circulating XBB.1.5 and XBB.1.16. Importantly, we found that both BQ.1 and XBB display an improved capacity to infect human nasal cells relative to ancestral D614G and/or Delta, like BA.1 does. However, compared to BA.1 and BQ.1, XBB exhibited a somewhat lower capacity to infect nasal ALI. These results can now be found in Figure 2E and Supplemental Figure 2C.

Reviewer #3 (Remarks to the Author):

Omicron Spike confers enhanced infectivity and interferon resistance to SARS-CoV-2 in human nasal tissue
Shi, Li et al.,

The authors present data showing the Omicron variant of SARS-CoV-2 increases infectivity in primary upper respiratory tract tissue. The entry of Omicron, unlike other variants, happens independently of TMPRSS2 and uses metalloproteinases to activate membrane fusion via the Spike. The authors show that Omicron is resistant to IFN-induced restriction following cell binding.

Comments:

1) The authors use Vero E6-TMPRSS2 cells to propagate the SARS-CoV-2 and then use Vero E6 cells to calculate infectious titers. The titers derived from Vero E6 infections are used to calculate the MOI for the infection studies using WA1, BA.1, BA.2. This approach may confound the MOI calculation as the Omicron variant is less dependent on TMPRSS2 and cathepsins for infectivity (Figure 2G). The authors should titer the viruses using Vero E6-TMPRSS2 to confirm the titres are not affected by the absence/presence of TMPRSS2 in Omicron.

We thank the reviewer for raising this good point. Actually, we titered all viruses on Vero E6-TMPRSS2 cells, not Vero E6 cells. It was a typographical error to state that viruses were titered on Vero E6 cells, so we apologize for this oversight. We have now corrected this mistake in the main text, methods section, and figure captions.

Additionally, as you will read below, we also titered viruses with viral qPCR as an alternative means to infer virus quantity. We found that using viral qPCR to calculate virus quantities supports our conclusion that BA.1 exhibits superior infectivity relative to WA1 and Delta (Supplemental Figure 1G and Supplemental Figure 2E).

2) Figure 1E: Virus attachment to cells show a 30-fold higher value for BA.1 compared to WA1. This is puzzling. It indicates that BA.1 has 30-fold higher infectious particles in the inoculant. If this is true, the higher FFUs seen in BA.1 at 24hpi are not surprising.

We understand that the reviewer may have been skeptical of this result. We now provide new data showing that the inoculants of WA1 and BA.1 infect HEK293T-ACE2 and Vero E6-TMPRSS2 cells to a similar degree, indicating that the BA.1 inoculant does not contain 30-fold

more infectious particles. Instead, these data demonstrate that BA.1 Spike selectively enables increased adherence to and infection of nasal cells. In our initial submission, we showed that the inoculant of BA.1 infected primary lung cells to a lesser extent than WA1, and this result would not be expected if the inoculant of BA.1 simply contained a higher number of infectious virus particles. These data are now available in Supplemental Figure 1E and 1F. We respond to this point further below.

3) Related to 2), the authors should rule out that the data shown in Figure 1D, 2B-2D, 2F, 3C are not just a reflection of higher infectious input in the inoculant of BA.1 or BA.2.

We agree that it is important to rule out that differing inoculums were partially or fully responsible for the difference in infection outcome, and another reviewer also pointed this out. Originally, we titered all of our viruses on Vero E6-TMPRSS2 cells to obtain infectious titers (the only way to infer infectious viral particle quantity). Now we have added new results to Supplemental Figure 1 showing that the inoculants used for WA1 and BA.1 result in comparable infection of HEK293T-ACE2 cells (Supplemental Figure 1E) and VeroE6-TMPRSS2 cells (Supplemental Figure 1F), indicating that differences in virus input quantity cannot explain the enhanced ability of BA.1 to infect nasal cells. Also, we showed in our initial submission that BA.1 exhibits poor replication in primary lung cells compared to WA1, and this result would not be expected if the inoculant of BA.1 was simply higher than that of WA.1.

We also chose to titer our virus using an alternative method (viral qPCR) in addition to measuring infectious titers by focus forming units assay in Vero E6-TMPRSS2. When we reperformed a side by side infection of WA1 and BA.1 using new titers calculated by viral qPCR, we observed that BA.1 still infected nasal cells significantly better than WA1 (Supplemental Figure 1G). Additionally, we reperformed infections with recombinant WA1-mCherry virus encoding WA1 Spike, Delta Spike, or BA.1 Spike using new titers calculated by viral qPCR. The results that were identical to those obtained through the traditional titering method (infectious virus particle enumeration on Vero E6-TMPRSS2 cells) (Supplemental Figure 2D and 2E).

Overall, our data indicate that BA.1 Spike confers a selective infectivity advantage that is apparent in nasal cells but not apparent in other cell lines. Therefore, we are able to exclude that differences in the inoculants are responsible for the perceived increased in nasal cell infectivity of Omicron.

4) P6 line3-5

“It is possible that an improved capacity for Omicron Spike to bind heparan sulfate and neuropilin 1 may negatively interfere with processing by TMPRSS2 and steer processing towards MMPs such as MMP14 and MMP16.” It may be informative to test the neuropilin 1 antagonist (EG00229) on its effect on Omicron.

We thank the reviewer for this terrific suggestion. We obtained EG00229 and tested its effect on nasal air-liquid interface infection by WA1, Delta, and BA.1. We found that WA1, Delta, and BA.1 were inhibited by EG00229 to different extents. Importantly, the sensitivity of viruses to EG00229 did not match their ability to infect/replicate in nasal air-liquid interface cells. For example, WA1 was the most sensitive to EG00229 (inhibited 5-fold), Delta was entirely resistant, and BA.1 exhibited modest sensitivity. These results are now found in Supplemental Figure 3, and

they suggest that the extent of neuropilin-1 dependency does not explain the differential ability for WA1, Delta, and BA.1 to infect nasal epithelia. Therefore, it is possible that other attachment/entry factors are responsible for the improved capacity for BA.1 Spike to adhere and trigger fusion with nasal cells. We have updated the paragraph in question to include discussion of these results at line 371: “One possibility explaining the enhanced capacity for Omicron Spike to adhere to and enter nasal cells (either in the presence or absence of ciliated cells) is its increase in net overall positive charge compared to previous variants of concern⁵⁹. It has also been proposed that Omicron Spike may adhere more strongly to attachment factors present at the epithelial cell surface, including heparan sulfate^{60,61} and neuropilin-1⁶². In this study, sensitivity to neuropilin-1 inhibitor did not correlate with the nasal cell infectivity of WA1, Delta, or BA.1, suggesting that other cellular factors may play a role (Supplemental Figure 3). A significant aim of future research will be to establish whether mutations in Omicron Spike promote increased residence time at the cell surface and whether this influences its dependence on certain cellular proteases for cleavage and the triggering of fusion. It is possible that longer residency at the cell surface may negatively interfere with processing by TMPRSS2 and steer processing towards metalloproteinases.”

5) The authors postulate that IFN induced proteins fail to block Omicron due to Omicron’s ability to enter cells independent of TMPRSS2. Do they have an indication of what IFN induced proteins are being bypassed by Omicron?

We believe that fully addressing the IFN-induced genes responsible for restriction of WA1 and Delta (and which fail to block Omicron) would require a genetic screening approach. Given that we use entirely primary human tissue for our work, this is outside the scope of this study. However, we decided to take advantage of the fact that Amphotericin B was previously shown to counteract the antiviral properties of IFITM2 and IFITM3 in cells. We have now added new data showing that Amphotericin B pre-treatment boosted infection of nasal air-liquid interface cells by WA1(WA1 Spike) and WA1(Delta Spike) by approximately 7-fold, suggesting that IFITM proteins exhibit antiviral properties in this human primary tissue. Interestingly, infection by WA1(BA.1 Spike) was boosted to a lesser extent by Amphotericin B (less than 2-fold). We employed recombinant viruses encoding mCherry to assess virus infection at multiple time points to come to these conclusions. Collectively, these results suggest that BA.1 Spike confers virus with partial resistance to antiviral IFITM proteins in nasal tissue. Therefore, it appears that IFITM proteins play a role in the limited virus replication exhibited by WA1 and Delta in human nasal tissue, while BA.1 Spike evolved at least partial resistance to IFITM proteins. Moreover, the significance of these results can be appreciated further when one considers that the role of IFITM proteins during SARS-CoV-2 infection has been controversial—some studies have raised the possibility that endogenous IFITM proteins are dependency factors for SARS-CoV-2 in certain cell lines (i.e. they are not antiviral). Our data demonstrate that endogenous IFITM proteins most likely perform inhibitory roles against SARS-CoV-2 in primary human nasal tissue and indicate that resistance to IFITM proteins may have contributed to the dominance of Omicron lineages in human populations. These results can now be found in Figure 5 and Supplemental Figure 5.

Minor points:

1) Does Omicron overcome restriction in cells over-expressing IFITMs?

We thank the reviewer for this suggestion. Rather than overexpress IFITM proteins in our primary tissue models, we decided to take address the impact of Amphotericin B treatment on infections by WA1 (WA1 Spike), WA1 (Delta Spike), and WA1 (BA.1 Spike) in primary nasal air-liquid interface tissue, as described in detail above. Our results can now be found in Figure 5 and Supplemental Figure 5, and they support the conclusion that Omicron evades restriction by IFITMs.

Typos:

P2 line3 – characterization of Omicron....

P5 last line and elsewhere in text (P6 line4) – neuropilin 1 not neurolipin 1.

These typos have been corrected. We thank the reviewer for pointing them out.

REVIEWER COMMENTS

Reviewer #1 (Remarks to the Author):

The resubmitted manuscript “Omicron Spike confers enhanced infectivity and interferon resistance to SARS-CoV-2 in human nasal tissue” by Shi and colleagues aims to understand contributing factors to increased Omicron BA.1 transmissibility compared to ancestral SARS-CoV-2 variants. As discussed in the original review there is interest in understanding of the global success of the Omicron variants, beyond antibody escape. While the revised manuscript has made efforts to address my comments, major shortcomings remain.

1 Addressing the issue of normalization of inoculum and dose for infection experiments. It is essential that the same amount of each variant is added to nasal cells in order to show differential infection/replication between Omicron and other variants, especially given that Omicron’s altered entry route/tropism is now well established. Here the authors have dosed input based on MOI calculations from titrations on Vero A/T cells. However it is critical to show that this equates to equal input using an alternate method that is agnostic of infection/tropism, hence the request to determine dose by measuring copies of viral Orf1a in the inoculum that was used to infect nasal cells. However, the new data presented in Fig S1E-G does not do this. Instead it shows measurements of Orf1a after 24h infection where the initial infections were still dosed by MOI. FigS1G in particular measures Orf1a at 24h post infection of nasal cells, but not an Orf1a measurement of the viral stocks used to infect the nasal cells, which was what was requested. Simply showing Orf1a is higher after 24h infection by BA1 vs WA1 could still be skewed by the MOI calculation meaning more BA1 is added to the nasal cultures initially. A higher BA1 input would also explain the data in Fig S1D where more BA1 is reported to attach to cells after 1h at 4C, i.e. more virus is added initially so more attaches and thus the “replication” after 24h is also higher. Careful measurements of the actual amount of Orf1a in the initial inoculum used to infect the primary cells for each variant should be performed and the comparison to MOI titre shown. I trust the authors have not exhausted all their initial virus stocks so this straightforward analysis could have been done.

2 The issue surrounding normalization of the data to a particular timepoint remains. The authors response is as follows “In order to render our normalized data more transparent, we now include infection results where WA1 and BA.1 were added to nasal monolayers or ALI and the absolute copy number of ORF1a was calculated. These results are now found in Supplemental Figure 1B and Supplemental Figure 2A, and the data demonstrate that BA.1 infection results in ORF1a copy numbers that are approximately two orders of magnitude greater than WA1”. The authors' make the point in their rebuttal that “It would be too resource intensive to reperform all of our infections so that we could replot all of our data in terms of ORF1a copy number. We believe that providing these examples documenting non-normalized ORF1a make all of the rest of the figures interpretable as normalized

graphs, since BA.1 is found in every figure where two or more variants were compared with one another”.

This reviewer did not ask for the authors to repeat all experiments, technically all of this is measurable in the experiments that they have done and does not require them to repeat all their infections. This reviewer requested the authors simply show/represent their data not normalized to 1 (i.e. absolute not relative values). The point of not normalizing the data is critical to discern the absolute levels of replication, but the vast majority of the data in the resubmission remains heavily normalized. The comment by this reviewer in the first round of reviews therefore still stands: Since this manuscript aims to understand differences between variants, normalizing infection levels within each variant to 1 does not allow direct comparisons between variants, nor allow the reader to understand the magnitude of changes in replication or IFN sensitivity that are reported.

3 Addressing the comment regarding differences in ACE2 expression levels on the primary airway cells used in the study, the authors now include some immunofluorescence images as a proxy to measure expression (Fig 2A). Unfortunately the methods states this is intracellular staining of ACE2 in permeabilized cells, thus it does not adequately address the question regarding ACE2 expression on the surface of cells, which would influence infection levels. The crucial data needed is to show surface staining of ACE2 on these cells, not intracellular ACE2. Moreover, there is no quantification of the images in Fig2A, so in addition to the fact that these data do not address surface expression, they cannot be interpreted.

4 Regarding the comments on the use of inhibitors to map Omicron entry pathway (Fig 3) I do not find the data in Fig 3A particularly convincing. The issue of specificity of the drugs for MMP inhibition at the doses used is not clearly addressed, and no drug titrations are shown. As far as I can tell, there has been no attempt to address this besides saying others used the drugs at this concentration so we did too. The authors do include new data with apratastat and batimastat to address MMPs/ADAMs usage (Fig 3C) but the effect of these inhibitors on BA1 infection is much reduced compared to Incyclinide. Thus I am not entirely convinced by the explanation that The fact that apratastat and batimastat inhibit BA.1 but not Delta indicates that they inhibit a narrower range of MMPs/ADAMs than incyclinide, and we therefore uncovered a functional distinction between the Delta and BA.1 entry pathways based on this result. What is the functional distinction? Weaker inhibitor may simply reflect incomplete drug activity due to the dose selected to use, but without careful titration this is impossible to know. Inhibitors are notoriously messy experiments that can be non-specific and without a proper titration (or control for activity) it is hard to interpret these data. Regarding the conclusion that “Given that some published studies and preprints claim that BA.1 enters human cells through an endocytic route, our findings showing that BA.1 does not enter via an endocytic route in primary nasal epithelia are of high significance to the scientific and lay communities, and we believe this is one of the reasons why our

study should be considered at Nature Communications” this is intriguing but again more work here is needed to conclusively show this.

5 To further address the question of differential replication of the variants and in an attempt to provide more supporting data, the authors “decided to expand the results obtained using our novel, recombinant WA1-mCherry viruses encoding WA1 Spike, Delta Spike, or BA.1 Spike for this purpose”. These new data raise more questions. The IF images of infection with these viruses are very poor quality, Fig S2F is impossible to interpret - I cannot see any staining at all for Delta spike infection over the 12 hours, and while there appears some signal for BA1 spike at 9 and 12hpi, these are low resolution images and again there is no quantification, so it impossible to interpret the magnitude of the effect size.

6 The authors use Amphotericin B to link the restriction phenotype to IFITMs to address what the cellular factor/ISGs responsible for the inhibition is. This plays to the Compton lab expertise in IFITM restriction factors. Indeed there is much literature on IFITM inhibition of SARS-CoV-2 which makes the choice obvious to follow up, but also reduces the novelty.

Fig 5 attempts to link the differential between the variants to IFITM restriction by using Amphotericin B to manipulate IFITMs. Unfortunately the data in Fig5 is again very hard to interpret. I don’t know what the images actually are. Are these 3D reconstructions? This is not explained and the images are poor quality and the quantification below lacks error bars, has no statistical analysis and the labels denoting which graph is which virus are not shown. If I assume correctly then the fluorescence signal in untreated BA1 spike is high and saturated and no difference to Ampho B is seen using a “relative fluorescence” but with such high levels of fluorescence I do not see how any increase would be evident, ie. the signal is already maxed out.

From these data the authors conclude that infection of nasal ALI cells by WA1 and Delta are more sensitive to IFITMs whereas BA1 spike has evolved partial resistance to IFITMs. This is intriguing and at odds with published work of others demonstrating that Omicron is sensitive to IFITM restriction – the difference between their work and that of others is surprisingly not discussed and no explanations are offered. One wonders how differences in expression levels of IFITMs across cell types and tissues may influence the results?

7 The inclusion of new data, as requested, using variants that contain the infection-enhancing D614G mutation is a useful addition and I am satisfied the authors addressed my concern about comparing Omicron that contains D614G with WA1 that does not from the original submission.

8 In the introduction, "...BA.1 contained a plethora of unique mutations, and its descendants (including the BA.2, BA.4, BA.5, BQ.1, and XBB subvariants)..." needs to be corrected BA.2 etc are not derived from BA.1, they share a common ancestor.

We thank the reviewer for pointing this out, and it has been corrected.

This has not been corrected.

Reviewer #2 (Remarks to the Author):

The authors have addressed all my points and provide new data that strengthen their conclusions. I have no further comments and support acceptance of the manuscript.

Reviewer #3 (Remarks to the Author):

The authors have adequately addressed my concerns in the revised manuscript. Thank you.

REVIEWER COMMENTS

Reviewer #1 (Remarks to the Author):

The resubmitted manuscript “Omicron Spike confers enhanced infectivity and interferon resistance to SARS-CoV-2 in human nasal tissue” by Shi and colleagues aims to understand contributing factors to increased Omicron BA.1 transmissibility compared to ancestral SARS-CoV-2 variants. As discussed in the original review there is interest in understanding of the global success of the Omicron variants, beyond antibody escape. While the revised manuscript has made efforts to address my comments, major shortcomings remain.

1 Addressing the issue of normalization of inoculum and dose for infection experiments. It is essential that the same amount of each variant is added to nasal cells in order to show differential infection/replication between Omicron and other variants, especially given that Omicron’s altered entry route/tropism is now well established. Here the authors have dosed input based on MOI calculations from titrations on Vero A/T cells. However it is critical to show that this equates to equal input using an alternate method that is agnostic of infection/tropism, hence the request to determine dose by measuring copies of viral Orf1a in the inoculum that was used to infect nasal cells. However, the new data presented in Fig S1E-G does not do this. Instead it shows measurements of Orf1a after 24h infection where the initial infections were still dosed by MOI. FigS1G in particular measures Orf1a at 24h post infection of nasal cells, but not an Orf1a measurement of the viral stocks used to infect the nasal cells, which was what was requested. Simply showing Orf1a is higher after 24h infection by BA1 vs WA1 could still be skewed by the MOI calculation meaning more BA1 is added to the nasal cultures initially. A higher BA1 input would also explain the data in Fig S1D where more BA1 is reported to attach to cells after 1h at 4C, i.e. more virus is added initially so more attaches and thus the “replication” after 24h is also higher. Careful measurements of the actual amount of Orf1a in the initial inoculum used to infect the primary cells for each variant should be performed and the comparison to MOI titre shown. I trust the authors have not exhausted all their initial virus stocks so this straightforward analysis could have been done.

We apologize if we misunderstood the reviewer’s request. We had believed that we responded to the reviewer’s criticism by performing ORF1a RT-qPCR on our virus stocks and using the ORF1a levels as titers. We then performed new infections using those qPCR-based titers, and our conclusions remained the same: BA.1 exhibits a significantly superior capacity to infect human nasal cells relative to WA1. However, now the reviewer is asking for the ORF1a levels present within all the virus stocks and a comparison with the titers calculated according to infection in Vero-ACE2-TMPRSS2 cells. We now list the ORF1a levels (ORF1a copies per 250 uL) and the infectious titers in Vero-ACE2-TMPRSS2 cells (focus-forming units per 250 uL), as well as the ORF1a copy number present in the virus inocula that were calculated according to the focus-forming units titer, in a new table in **Supplemental Figure 8**. Our results indicate that the amount of ORF1a RNA in the virus inoculum is not a good predictor of infection outcome at 24 or 48 hours post-inoculation. For example, the ORF1a RNA levels in the BA.1 inoculum are about 6-fold greater than those of WA1, but BA.1 achieves infection yields that are more than 250-fold greater (Figure 2E). Even if you consider the attachment assay results, BA.1 displayed a more than 30-fold increased adherence to nasal cells (Supplemental Figure 2D). A 30-fold

difference cannot result from a 6-fold difference in ORF1a RNA present in the inocula. Moreover, the ORF1a RNA levels in the BQ.1 inoculum are 2-fold greater than those of WA1, yet BQ.1 reaches infection yields that are about 100-fold greater (Figure 2E). Furthermore, the inoculum of XBB contained more ORF1a RNA than the inoculum of BA.1, yet BA.1 achieved higher infection yields than XBB (Supplemental Figure 3C). Lastly, the ORF1a levels present in the inocula of the mCherry recombinant viruses differ by no more than 2-fold, yet WA1 (BA.1 Spike) exhibits an infectivity that is 100-fold greater than WA1 (WA1 Spike) and WA1 (Delta Spike) (Supplemental Figure 3E). Overall, these results demonstrate that differences in ORF1a RNA present in viral inocula do not impact our ability to conclude that Omicron lineages display enhanced infectivity in nasal epithelia relative to previous virus variants.

2 The issue surrounding normalization of the data to a particular timepoint remains. The authors response is as follows “In order to render our normalized data more transparent, we now include infection results where WA1 and BA.1 were added to nasal monolayers or ALI and the absolute copy number of ORF1a was calculated. These results are now found in Supplemental Figure 1B and Supplemental Figure 2A, and the data demonstrate that BA.1 infection results in ORF1a copy numbers that are approximately two orders of magnitude greater than WA1”. The authors' make the point in their rebuttal that “It would be too resource intensive to reperform all of our infections so that we could replot all of our data in terms of ORF1a copy number. We believe that providing these examples documenting non-normalized ORF1a make all of the rest of the figures interpretable as normalized graphs, since BA.1 is found in every figure where two or more variants were compared with one another”.

This reviewer did not ask for the authors to repeat all experiments, technically all of this is measurable in the experiments that they have done and does not require them to repeat all their infections. This reviewer requested the authors simply show/represent their data not normalized to 1 (i.e absolute not relative values). The point of not normalizing the data is critical to discern the absolute levels of replication, but the vast majority of the data in the resubmission remains heavily normalized. The comment by this reviewer in the first round of reviews therefore still stands: Since this manuscript aims to understand differences between variants, normalizing infection levels within each variant to 1 does not allow direct comparisons between variants, nor allow the reader to understand the magnitude of changes in replication or IFN sensitivity that are reported.

We understand that the reviewer continues to be frustrated by the presence of normalized data in our manuscript. So, we decided to replot all of our normalized data as non-normalized figures which can be found in a new **Supplemental Appendix PDF** file. Now, readers can consult the non-normalized data in the supplemental material if they wish. Furthermore, the new data introduced into the main figures now features quantitative data that is non-normalized (**Figure 5C**, the quantitative measurement of mCherry fluorescence from recombinant viruses in untreated and Amphotericin B-treated nasal ALI).

3 Addressing the comment regarding differences in ACE2 expression levels on the primary airway cells used in the study, the authors now include some immunofluorescence images as a proxy to measure expression (Fig 2A). Unfortunately the methods states this is intracellular staining of ACE2 in permeabilized cells, thus it does not adequately address the question

regarding ACE2 expression on the surface of cells, which would influence infection levels. The crucial data needed is to show surface staining of ACE2 on these cells, not intracellular ACE2. Moreover, there is no quantification of the images in Fig2A, so in addition to the fact that these data do not address surface expression, they cannot be interpreted.

We understand the reviewer's insistence to provide some demonstration that the ACE2 protein detected in nasal monolayers can be observed at the cell surface. We now provide a new image in Figure 1A that shows ACE2 expression in nasal monolayers. In this improved example, ACE2 can be seen not only to intracellular compartments but also at the apparent cell surface (**Figure 1A**). Furthermore, we also performed additional staining experiments whereby we combined anti-ACE2 immunofluorescence with labeling of the cell surface with phalloidin. Phalloidin selectively marks filamentous actin at or very near the plasma membrane, and it is commonly used in confocal microscopy to label the cell surface of cells. Examples of ACE2 and phalloidin colocalization can now be seen in **Supplemental Figure 1**. We provide several slices from confocal stacked images to show that a portion of the ACE2 in nasal monolayer cells is at or very near the plasma membrane. We chose to adopt this approach to assess ACE2 at the cell surface because the method suggested by the reviewer (ACE2 staining of non-permeabilized cells) requires that the cells be alive during the staining procedure, and our lab has not had success with this approach in the past. Moreover, our approach indicates where exactly at the cell surface ACE2 is located (since the whole cell surface is lit up by phalloidin) and it enables the reader to visualize intracellular and cell surface ACE2 simultaneously. Regarding quantification of ACE2, we feel this is unnecessary because we are not comparing ACE2 signal intensity between two or more conditions, nor are we making conclusions on ACE2 quantities. Taken together with the data we provided in the first revision, showing that nasal monolayers are productively infected by SARS-CoV-2 variants and capable of producing infectious virions, there is no longer any reason to doubt that these nasal monolayers do not serve as a useful model for studying SARS-CoV-2 variants.

4 Regarding the comments on the use of inhibitors to map Omicron entry pathway (Fig 3) I do not find the data in Fig 3A particularly convincing. The issue of specificity of the drugs for MMP inhibition at the doses used is not clearly addressed, and no drug titrations are shown. As far as I can tell, there has been no attempt to address this besides saying others used the drugs at this concentration so we did too. The authors do include new data with apratastat and batimastat to address MMPs/ADAMs usage (Fig 3C) but the effect of these inhibitors on BA1 infection is much reduced compared to Incyclinide. Thus I am not entirely convinced by the explanation that The fact that apratastat and batimastat inhibit BA.1 but not Delta indicates that they inhibit a narrower range of MMPs/ADAMs than incyclinide, and we therefore uncovered a functional distinction between the Delta and BA.1 entry pathways based on this result. What is the functional distinction? Weaker inhibitor may simply reflect incomplete drug activity due to the dose selected to use, but without careful titration this is impossible to know. Inhibitors are notoriously messy experiments that can be non-specific and without a proper titration (or control for activity) it is hard to interpret these data. Regarding the conclusion that "Given that some published studies and preprints claim that BA.1 enters human cells through an endocytic route, our findings showing that BA.1 does not enter via an endocytic route in primary nasal epithelia are of high significance to the scientific and lay communities, and we believe this is one of the reasons why our study should be considered at Nature Communications" this is intriguing but

again more work here is needed to conclusively show this.

We understand that it is extremely difficult to interpret data generated by inhibitors, much less multiple inhibitors compared to one another. For that reason, we have altered the language in our Results section to tone down our conclusions and to avoid overstating the results. What we think is clear, though, is that BA.1 is more sensitive to all MMP inhibitors tested compared to Delta. We have removed the line referenced by the reviewer containing the words “functional distinction” between Delta and BA.1 Spike and the line suggesting that BA.1 uses ADAM10 and ADAM17 for entry. At line 221, we now state “While incyclinide inhibited WA1 (Delta Spike) infection, apratastat and batimastat did not inhibit but instead modestly boosted infection (Figure 3C). In contrast, both apratastat and batimastat reduced WA (BA.1 Spike) infection by 60%. These results may suggest that, compared to Delta Spike, BA.1 Spike is more dependent on processing by metalloproteinases. Therefore, BA.1 Spike has evolved to utilize a distinct entry route in nasal epithelia that is TMPRSS2-independent and metalloproteinase-dependent.” Note that we employ the term “may” to exercise caution and indicate uncertainty. Furthermore, we have also toned down our statements on BA.1 using a non-endocytic entry pathway. We removed “independent of endocytosis” and “non-endocytic” from the subtitles and figure legend captions.

5 To further address the question of differential replication of the variants and in an attempt to provide more supporting data, the authors “decided to expand the results obtained using our novel, recombinant WA1-mCherry viruses encoding WA1 Spike, Delta Spike, or BA.1 Spike for this purpose”. These new data raise more questions. The IF images of infection with these viruses are very poor quality, Fig S2F is impossible to interpret - I cannot see any staining at all for Delta spike infection over the 12 hours, and while there appears some signal for BA1 spike at 9 and 12hpi, these are low resolution images and again there is no quantification, so it impossible to interpret the magnitude of the effect size.

We are sorry that the reviewer was frustrated by our IF images, but we believe it is due to the fact that the images were small in order to fit within the confines of the figures. The Supplemental Figure 2F that is referred to has now been redrawn as **Supplemental Figure 4**, with increased image size and higher resolution. It should now be very clear that WA1 (BA.1 Spike) becomes detectable as early as 6 hours post inoculation. We also add quantification (non-normalized) of the images in **Supplemental Figure 4B** to indicate the effect size. The modest amount of mCherry signal in the WA1 (Delta Spike) condition at 12 hours post-inoculation should now be apparent to the reviewer. The amount of infection by WA1 (Delta Spike) at different time points is now quantified and can be compared to that of WA1 (BA.1 Spike). We believe the readers can now fully interpret and appreciate these results, which reinforce that BA.1 Spike confers enhanced and rapid spread through the nasal epithelia.

6 The authors use Amphotericin B to link the restriction phenotype to IFITMs to address what the cellular factor/ISGs responsible for the inhibition is. This plays to the Compton lab expertise in IFITM restriction factors. Indeed there is much literature on IFITM inhibition of SARS-CoV-2 which makes the choice obvious to follow up, but also reduces the novelty.

Fig 5 attempts to link the differential between the variants to IFITM restriction by using Amphotericin B to manipulate IFITMs. Unfortunately the data in Fig5 is again very hard to

interpret. I don't know what the images actually are. Are these 3D reconstructions? This is not explained and the images are poor quality and the quantification below lacks error bars, has no statistical analysis and the labels denoting which graph is which virus are not shown. If I assume correctly then the fluorescence signal in untreated BA1 spike is high and saturated and no difference to Ampho B is seen using a "relative fluorescence" but with such high levels of fluorescence I do not see how any increase would be evident, ie. the signal is already maxed out.

We are sorry that the reviewer had trouble interpreting this data. The images in Figure 5 are not 3D reconstructions, but rather, images of the entire well of tissue, rather than a higher magnification of the tissue in part of the well (like what appears in Figure 2G). We have added this detail to the figure caption to avoid confusion. To increase the interpretability of the data, we have also added quantification (non-normalized) of the mCherry fluorescence in each condition and included an additional repetition of the experiment. Now, the fluorescence is quantified, and symbols and error bars indicate the variance in the data. Since the data is no longer normalized, we removed the term "relative fluorescence." We increased the size and resolution of the images showing the impact of Ampho B on WA1 (BA.1 Spike) at 9 and 12 hours post-inoculation, now seen in **Supplemental Figure 7**. Regarding the signal being potentially maxed out in the WA1 (BA.1 Spike) infected tissue, we now provide more new data in **Supplemental Figure 7** showing that the mCherry fluorescence signal becomes saturated at 36 hours post-inoculation, but not at 12 hours or 24 hours post-inoculation. Therefore, we are confident of our mCherry fluorescence quantifications and the conclusions derived from these experiments.

From these data the authors conclude that infection of nasal ALI cells by WA1 and Delta are more sensitive to IFITMs whereas BA1 spike has evolved partial resistance to IFITMs. This is intriguing and at odds with published work of others demonstrating that Omicron is sensitive to IFITM restriction – the difference between their work and that of others is surprisingly not discussed and no explanations are offered. One wonders how differences in expression levels of IFITMs across cell types and tissues may influence the results?

We appreciate this comment by the reviewer and agree that we should reference recently published work characterizing the impact of IFITM on Omicron infection. We now cite two papers whose results conflict with one another. Mesner et al. PNAS, 2023 showed that overexpressed IFITM proteins inhibit Omicron in transformed cell lines, and to a greater extent than previous variants of concern. Nchioua et al. JVI, 2022 showed that human IFITM proteins are important dependency factors for SARS-CoV-2 variants in lung cells. These data are not in agreement with our finding (for different reasons) that BA.1 evades restriction by IFITM protein in primary nasal tissue. We believe that multiple reasons may underlie the different conclusions presented by these papers. We now state at line 399 of the Discussion that "It is worth noting that our results showing that BA.1 Spike-mediated infection is resistant to the antiviral activities of IFITM proteins in nasal epithelia is not in agreement with recent publications. It is possible that the impacts of IFITM proteins on SARS-CoV-2 infection may vary depending on cell type (transformed versus primary), tissue type (lung versus nasal), and virus type (pseudovirus versus replication-competent clinical isolates)."

7 The inclusion of new data, as requested, using variants that contain the infection-enhancing

D614G mutation is a useful addition and I am satisfied the authors addressed my concern about comparing Omicron that contains D614G with WA1 that does not from the original submission.

Thank you.

8 In the introduction, "...BA.1 contained a plethora of unique mutations, and its descendants (including the BA.2, BA.4, BA.5, BQ.1, and XBB subvariants)..." needs to be corrected BA.2 etc are not derived from BA.1, they share a common ancestor.

We thank the reviewer for pointing this out, and it has been corrected.

This has not been corrected.

It is now corrected. Sorry for the confusion. At line 47, the sentence now reads "Compared to previous SARS-CoV-2 variants of concern, Spike from BA.1 contained a plethora of unique mutations, and subsequent Omicron lineages (including the BA.2, BA.4, BA.5, BQ.1, and XBB subvariants) contained additional mutations in Spike."

Reviewer #2 (Remarks to the Author):

The authors have addressed all my points and provide new data that strengthen their conclusions. I have no further comments and support acceptance of the manuscript.

Reviewer #3 (Remarks to the Author):

The authors have adequately addressed my concerns in the revised manuscript. Thank you.

REVIEWERS' COMMENTS

Reviewer #1 (Remarks to the Author):

Reviewer comments:

1. The addition of Supplemental Figure 8 is useful and allows the readers to make up their own mind about whether the experiments are well-controlled for dosing with viral inoculum and the subsequent replication differences observed. However, I note that Supplemental Figure 8 is not referred to in the manuscript. I trust this is simply an oversight (?) but given the extended discussion about normalization of dose and the concerns of this reviewer, Supplemental Figure 8 must be referred to directly and the results text should include statement contrasting the different measurements (FFU vs gRNA/particle) to equalise input (i.e. that Orf1a RNA levels vary strikingly when compared to FFUs).

2. The inclusion of most of the requested non-normalized data is helpful and again allows the readers to make up their own mind about the effect sizes.

3. Regarding ACE2 expression, the new images are improved and helpful for the reader. I do not agree with the authors that it is not necessary to quantify immunofluorescence images but if they do not wish to include that then again, the readers can interpret the data as they see fit.

4. The addition of text in the discussion referencing how the data in this manuscript relate to other published work on the impact of IFITM on Omicron infection is welcomed.

5. Regarding the inhibitor data, the manuscript text has not been updated as stated in the response and this needs correcting. See below and note absence of the word may “these results may suggest that” in rebuttal vs “these results suggest that” in the manuscript.

Rebuttal letter:

“While incyclinide inhibited WA1 (Delta Spike) infection, apratastat and batimastat did not inhibit but instead modestly boosted infection (Figure 3C). In contrast, both apratastat and batimastat reduced WA (BA.1 Spike) infection by 60%. These results may suggest that, compared to Delta Spike, BA.1 Spike is more dependent on processing by metalloproteinases. Therefore, BA.1 Spike has evolved to utilize a distinct entry route in nasal epithelia that is TMPRSS2-independent and metalloproteinase-dependent.” Note that we employ the term “may” to exercise caution and indicate uncertainty. Furthermore, we have

also toned down our statements on BA.1 using a non-endocytic entry pathway. We removed “independent of endocytosis” and “non-endocytic” from the subtitles and figure legend captions.

Manuscript text:

“While incyclinide inhibited WA1 (Delta Spike) infection, apratastat and batimastat did not inhibit but instead modestly boosted infection (Figure 3C). In contrast, both apratastat and batimastat reduced WA (BA.1 Spike) infection by 60% (Figure 3C). These results suggest that, compared to Delta Spike, BA.1 Spike is more dependent on processing by metalloproteinases. Therefore, BA.1 Spike has evolved to utilize a distinct entry route in nasal epithelia that is TMPRSS2-independent and metalloproteinase-dependent”.

REVIEWERS' COMMENTS

Reviewer #1 (Remarks to the Author):

Reviewer comments:

1. The addition of Supplemental Figure 8 is useful and allows the readers to make up their own mind about whether the experiments are well-controlled for dosing with viral inoculum and the subsequent replication differences observed. However, I note that Supplemental Figure 8 is not referred to in the manuscript. I trust this is simply an oversight (?) but given the extended discussion about normalization of dose and the concerns of this reviewer, Supplemental Figure 8 must be referred to directly and the results text should include statement contrasting the different measurements (FFU vs gRNA/particle) to equalise input (i.e. that Orf1a RNA levels vary strikingly when compared to FFUs).

We thank the reviewer for bringing this oversight to our attention. Indeed, it was our intention to call out this new Supplemental Figure in the Results section. We renamed this Figure to Supplemental Table 1 in order to meet the formatting requirements of the journal. We now call out Supplemental Table 1 at line 106 of the Results section: “Since our virus stocks contained varying concentrations of viral RNA (Supplemental Table 1), we excluded that WA1 and BA.1 inoculants (initially titered on Vero E6-ACE2-TMPRSS2 cells) contained different quantities of virus particles by inoculating HEK293T-ACE2 and Vero E6-ACE2-TMPRSS2 cells in parallel.”

2. The inclusion of most of the requested non-normalized data is helpful and again allows the readers to make up their own mind about the effect sizes.

Thank you.

3. Regarding ACE2 expression, the new images are improved and helpful for the reader. I do not agree with the authors that it is not necessary to quantify immunofluorescence images but if they do not wish to include that then again, the readers can interpret the data as they see fit.

Thank you.

4. The addition of text in the discussion referencing how the data in this manuscript relate to other published work on the impact of IFITM on Omicron infection is welcomed.

Thank you.

5. Regarding the inhibitor data, the manuscript text has not been updated as stated in the response and this needs correcting. See below and note absence of the word may “these results may suggest that” in rebuttal vs “these results suggest that” in the manuscript.

Rebuttal letter:

“While incyclinide inhibited WA1 (Delta Spike) infection, apratastat and batimastat did not inhibit but instead modestly boosted infection (Figure 3C). In contrast, both apratastat and batimastat reduced WA (BA.1 Spike) infection by 60%. These results may suggest that,

compared to Delta Spike, BA.1 Spike is more dependent on processing by metalloproteinases. Therefore, BA.1 Spike has evolved to utilize a distinct entry route in nasal epithelia that is TMPRSS2-independent and metalloproteinase-dependent.” Note that we employ the term “may” to exercise caution and indicate uncertainty. Furthermore, we have also toned down our statements on BA.1 using a non-endocytic entry pathway. We removed “independent of endocytosis” and “non-endocytic” from the subtitles and figure legend captions.

Manuscript text:

“While incyclinide inhibited WA1 (Delta Spike) infection, apratastat and batimastat did not inhibit but instead modestly boosted infection (Figure 3C). In contrast, both apratastat and batimastat reduced WA (BA.1 Spike) infection by 60% (Figure 3C). These results suggest that, compared to Delta Spike, BA.1 Spike is more dependent on processing by metalloproteinases. Therefore, BA.1 Spike has evolved to utilize a distinct entry route in nasal epithelia that is TMPRSS2-independent and metalloproteinase-dependent“.

We thank the reviewer for pointing out this oversight. We have now corrected this by introducing the word “may” before “suggest that.” Now, at line 225, the line reads “These results may suggest that, compared to Delta Spike, BA.1 Spike is more dependent on processing by metalloproteinases. Therefore, BA.1 Spike has evolved to utilize a distinct entry route in nasal epithelia that is TMPRSS2-independent and metalloproteinase-dependent.”